# Structural features discriminating hybrid histidine kinase Rec domains from response regulator homologs

Mitchell Brüderlin[1], Raphael Böhm[1], Firas Fadel[1], Sebastian Hiller[1], Tilman Schirmer[1] ✉ & Badri N. Dubey[1,2] ✉

In two-component systems, the information gathered by histidine kinases (HKs) are relayed to cognate response regulators (RRs). Thereby, the phosphoryl group of the auto-phosphorylated HK is transferred to the receiver (Rec) domain of the RR to allosterically activate its effector domain. In contrast, multi-step phosphorelays comprise at least one additional Rec ($Rec_{inter}$) domain that is typically part of the HK and acts as an intermediary for phosphoryl-shuttling. While RR Rec domains have been studied extensively, little is known about discriminating features of $Rec_{inter}$ domains. Here we study the $Rec_{inter}$ domain of the hybrid HK CckA by X-ray crystallography and NMR spectroscopy. Strikingly, all active site residues of the canonical Rec-fold are pre-arranged for phosphoryl-binding and $BeF_3^-$ binding does not alter secondary or quaternary structure, indicating the absence of allosteric changes, the hallmark of RRs. Based on sequence-covariation and modeling, we analyze the intra-molecular DHp/Rec association in hybrid HKs.

Kinases constitute central cellular switches in all domains of life. Precise control of their activity in time and space is vital for the implementation of specific cellular programs. Bacteria sense and respond to a wide variety of signals through a complex network of signaling systems such as two-component system (TCS) pathways. TCSs constitute the major signal transduction system to regulate vital processes such as chemotaxis, cell cycle, virulence, etc., in response to external and endogenous stimuli. A chemical or physical input signal is perceived by the input domain of a sensory histidine kinase (HK), causing auto-phosphorylation of its conserved histidine on the DHp transmitter domain (Fig. 1a). This is followed by phosphotransfer to the conserved aspartate of the receiver (Rec) domain of a cognate response regulator (RR) to activate its effector domain to finally elicit the output[1]. Typically, effector domains regulate transcription (e.g., PhoB, OmpR) or enzymatic activity (CheB, HK, PleD), for a review see Galperin, 2010[2]. Rec domains of transcriptional regulators have been studied extensively. In the OmpR family, phosphorylation-induced Rec dimerization mediated by Y/T coupling enables DNA binding and, thus, transcriptional control[1]. For the Rec-controlled diguanylate cyclase DgcR, a

change in the preformed dimeric coiled-coil association of the C-terminal Rec helices has been reported recently as the basis of activation[3].

Multi-step phosphorelays are more complex involving at least one additional Rec domain and a histidine-containing phosphotransferase (either an HPt protein or a pseudo-HK), both of which act as intermediaries in the phosphotransfer[1,4]. Often, the additional Rec domain is fused C-terminally to the HK to form a hybrid histidine kinase (HHK) (Fig. 1b). As a well-studied example, the HHK CckA initiates the *Caulobacter crescentus* CckA–ChpT–CtrA/CpdR phosphorelay (Fig. 1c) which is central to the regulation of the cell cycle via the transcription master regulator CtrA[5]. As endogenous input signals, DivL[6,7] and the second messenger c-di-GMP[8], the concentration of which varies over the cell cycle, have been identified[9]. Multi-step phosphorelays comprise two types of Rec domains that can be distinguished functionally: (1) Rec domains that act as intermediaries in the phosphotransfer chain by shuttling the phosphoryl between the active histidines of an HHK and a histidine phosphotransferase (hereafter referred to as $Rec_{inter}$) and (2) Rec domains that constitute the terminal phosphoryl acceptors

[1]Structural Biology, Biozentrum, University of Basel, Spitalstr. 41, 4056 Basel, Switzerland. [2]CSSB Centre for Structural Systems Biology, Deutsches Elektronen-Synchrotron DESY, Notkestr. 85, 22607 Hamburg, Germany. ✉e-mail: tilman.schirmer@unibas.ch; badri.nath.dubey@desy.de

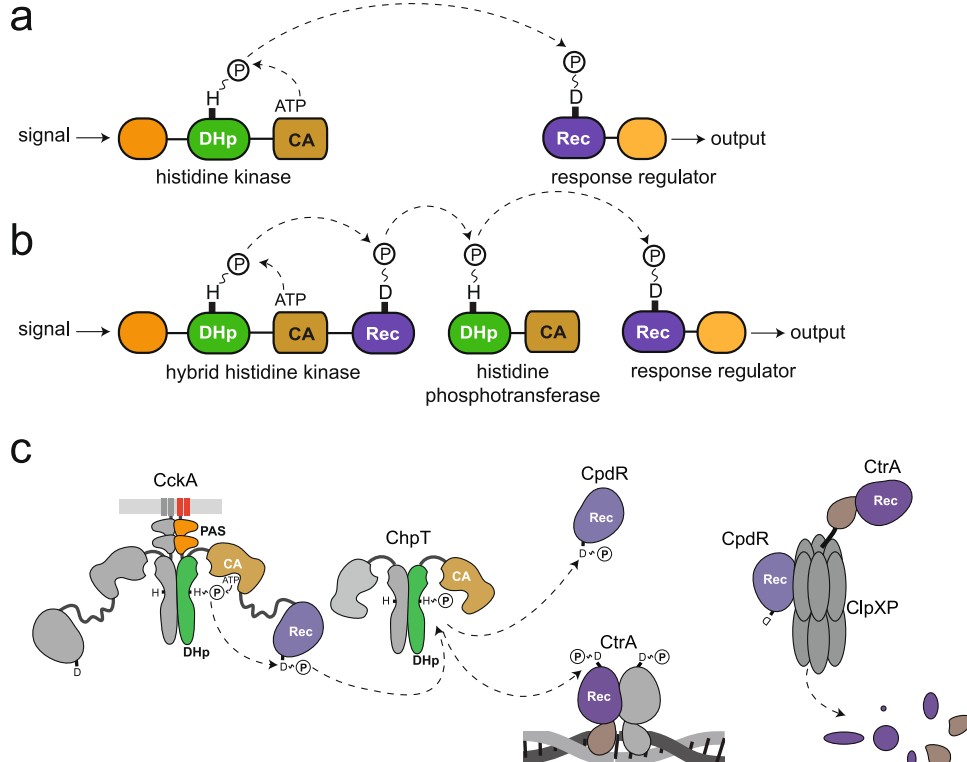

**Fig. 1 | Schematic representation of two-component vs multi-component signaling pathways and the CckA–ChpT–CtrA/CpdR phosphorelay system.**
**a** Classical two-component system comprised of a histidine kinase and a response regulator. Domain organisation and residues involved in phosphoryl transfer are indicated. **b** Multistep phosphorelay consisting of a hybrid histidine kinase (HHK), a histidine phosphotransferase, and a response regulator. The histidine phosphotransferase can either be an HPt protein or a pseudo-HK, as shown.
**c** CckA–ChpT–CtrA/CpdR phosphorelay of *Caulobacter crescentus*. Each subunit of homo-dimeric HHK CckA (left) is comprised of two transmembrane helices (red),

two Per-Arnt-Sim (PAS) domains (orange), a dimerization/histidine phospho-transfer (DHp, green) domain, a catalytic ATP binding (CA, beige) domain and a C-terminal Rec (Rec, violet) domain. The domains of ChpT are structurally related to the DHp and CA domains of CckA, although the latter is functionally degenerated. ChpT can phosphorylate two acceptors, the DNA-binding response regulator CtrA and the single-Rec protein CpdR. P~CtrA binds as a dimer to the origin of replication to inhibit replication initiation. In parallel, CpdR phosphorylation impedes binding of CpdR to the protease complex ClpXP, which would prime the protease for CtrA degradation.

(hereafter referred to as Rec$_{term}$) and typically control the activity of associated effector domains in RRs. Rec-only proteins can belong to either class with, e.g., MrrA shuttling phosphoryl groups between various kinases[10] and CpdR regulating the activity of a protease via complex formation[11] (Fig. 1c).

Rec domains of HHKs are of the Rec$_{inter}$ type by definition. It has been estimated that they make up about 10% of all known Rec domains[1], but only very little is known about their structural and functional properties[12,13]. Previously, it has been shown that Rec$_{inter}$ domains are not using the 4-5-5 face for interacting with their phosphorelay partners[14,15], therefore we hypothesized that the allosteric mechanism evolved in Rec$_{term}$ domains to change the structure of the 4-5-5 face would not be needed.

Here, to test for distinguishing features of Rec$_{inter}$ domains, we analyze the structure and dynamics of the C-terminal Rec domain of the well-studied HHK CckA (CckA$^{Rec}$) and find, quite unusual, its active site in the active conformation and indeed no allosteric response upon beryllofluoride modification. The structure of CckA$^{Rec}$ together with that of the previously determined structure of CckA$^{DHp-CA}$[16] allows us to model the intramolecular phosphotransfer competent conformation of CckA based on the structure of the ChpT/CtrA complex from *B. abortus*[14]. Most interface residues are not conserved but show very clear co-evolution amongst HHK sequences. This probably prevents cross-talk of the Rec domain with downstream partners of the various phosphorelays.

## Results

### Crystal structure of CckA$^{Rec}$
The crystal structure of the Rec domain of CckA (CckA$^{Rec}$, residues 568–691) was determined to very high resolution (1.25 Å resolution) by molecular replacement (Fig. 2a). Data collection and refinement statistics are given in Table 1. The structure is defined by continuous electron density from residues 570 to 689 except for residues 654 to 663, with some poor density but not clear enough to model the linker segment reliably (Supplementary Fig. 1). CckA$^{Rec}$ exhibits the canonical (β/α)$_5$ fold of Rec domains with a central five-stranded parallel β-sheet and the β-strands connected by cross-over helices[17]. However, the cross-over segment from β4 to β5 (β4–β5 linker) is partially disordered and not folded into a canonical α4 helix (Fig. 2a).

The structure of CckA$^{Rec}$ superimposes only moderately well (rmsd = 1.6 Å for 104 Cα positions) with the Rec domain of the CtrA homolog from *B. abortus* (Fig. 2a, b), although both domains are functionally engaged with the histidine phosphotransferase ChpT during phosphorelay (Fig. 1c). Rec structures with a disordered β4–β5 segment, i.e. a missing α4 helix, have been seen before, e.g., for non-activated PhoB (2iyn[18]) or for the stand-alone Rec domain Q1CZZ7_MYXXD (3nhm). A β4–β5 segment with an irregular loop structure is seen in the stand-alone Rec protein Mhun_0886 (3cg4, Fig. 2c). The cryptic Rec1 domain of the HHK ShkA involved in c-di-GMP binding shows an irregular β3–β4 linker (6qrl[19], Fig. 2d) and, in the Rec1 domain of HHK RcsC (2ayx[20]), three helices are replaced by irregular loops. Thus, the Rec-fold appears stable enough to tolerate

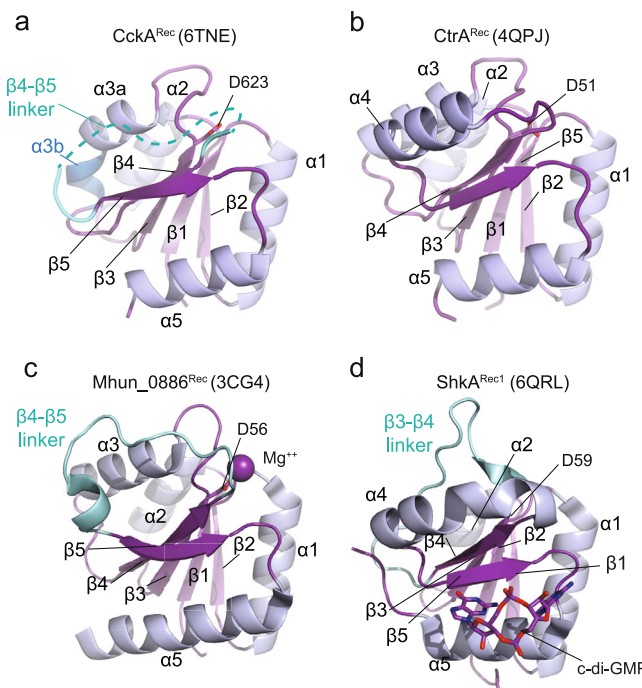

**Fig. 2 | Structure comparison of CckA^Rec with the canonical fold of CtrA^Rec, and with other Rec domains lacking secondary structure elements.** Cartoon representation with β-strands colored in purple and α-helices in light-blue. Non-canonical elements are indicated in cyan. PDB codes are given in brackets. **a** CckA^Rec displaying the classical (βα)₅, but with a disordered β4–β5 linker instead of an α4 helix. Note, that α3 is broken into two pieces (α3a and α3b) as indicated by the difference in coloring. **b** As reference, CtrA^Rec from *B. abortus* shows the canonical (βα)₅ Rec fold. **c** The β4–β5 linker of receiver domain Mhun_0886^Rec exhibits irregular loop structure with a small helical section. **d** The cryptic receiver domain ShkA^Rec1 lacks α3 and binds cyclic-di-GMP that modulates the function of the full-length protein.

such variations in elements that may be functionally dispensable or involved in the specialized function.

As anticipated from its canonical fold, the CckA^Rec structure aligns well with many Rec domains in the PDB database including prokaryotic as well as yeast and plant homologs. When ranked according to the PDBeFold Q score, the top hit is a structure of SpoOF from *Bacillus subtilis* (3q15[21], rmsd = 1.2 Å for 100 of the 109 CckA^Rec Cα positions) followed by two other stand-alone Rec proteins, DivK_CAUCR and Q1CZZ7_MYXXD.

The asymmetric unit of the CckA^Rec crystals contains one Rec monomer and no significant interfaces are formed within crystal lattice (all interface area <275 Å²). Indeed, as measured by SEC-MALS, CckA^Rec was found to be monomeric in solution up to concentrations of around 1 mM (Supplementary Fig. 2).

### Active site of CckA^Rec exhibits an activated conformation

Figure 3a and b show a close-up view of the phosphorylation site (D623) with surrounding residues and hydrating water molecules. Unexpectedly, the divalent cation binding site is not occupied, with the closest water (W739) at a distance of 1.7 Å from the site. K673 (from the β5 to α5 loop) implicated in phospho-stabilisation and the "switch" residues S651 (end of β4) and F670 (β5) are well defined by electron density with the latter residue exhibiting two alternative rotamer conformations.

Though the CckA^Rec structure has been determined in absence of the phosphoryl-mimic berylloflaoride and shows no bound divalent cation, the active site is clearly in the activated conformation. This can be inferred from, e.g., its comparison with the native and BeF₃⁻-

**Table 1 | Crystallographic data collection and refinement statistics**

| Data collection | |
| --- | --- |
| Synchrotron source | SLS, PXI |
| Wavelength (Å) | 1.00004 |
| Space group | I 1 2 1 |
| a, b, c (Å) | 37.4, 42.9, 65.0 |
| α, β, γ (°) | 90.0, 94.4, 90.0 |
| Resolution (Å) | 28.2-1.25 (1.29-1.25) |
| No. of unique reflections | 28065 (2818) |
| No of reflections used for R$_{free}$ | 1372 (136) |
| Completeness (%) | 98.37 (99.58) |
| I/σ (I) | 17.24 (4.96) |
| Redundancy | 2.0 (2.0) |
| R$_{merge}$ (%) | 1.7 (12.9) |
| R$_{pim}$ (%) | 1.7 (12.9) |
| CC (1/2) (%) | 99.9 (94.0) |
| **Refinement** | |
| R$_{work}$/R$_{free}$ (%) | 16.5/18.8 |
| RMSD from ideal values | |
| -- Bond length (Å) | 0.014 |
| -- Bond angles (°) | 1.45 |
| Molecules/asymmetric unit | 1 |
| No. of atoms | |
| -- Protein | 839 |
| -- Water | 103 |
| **Average B-factor (Å²)** | 18.9 |
| -- Protein (Å²) | 20.9 |
| -- Water (Å²) | 38.2 |
| **Ramachandran statistics** | |
| -- Favored regions (%) | 100.0 |
| -- Allowed regions (%) | 0.0 |
| -- Disallowed regions (%) | 0.0 |
| PDB Code | 6TNE |

modified active site structures of PhoB[22] (Supplementary Fig. 3a, b). Like activated PhoB^Rec, CckA^Rec shows (1) a closed β3–α3 loop with main-chain carbonyl 625 (55 in PhoB^Rec) in place for Mg⁺⁺ coordination and (2) the side-chain hydroxyl of S651 (T83 in PhoB^Rec) in the inward position ready for forming an H-bond with an incoming BeF₃⁻ - moiety (Fig. 3c, d). We note, that one of the two alternative conformations of the aromatic switch residue F670 coincides with the inward orientation as seen in activated PhoB (Fig. 3d).

### Secondary structure and dynamics of native and BeF₃⁻ modified CckA^Rec in solution

To investigate the effect of activation on the structure and dynamics of CckA^Rec, NMR studies were performed on native and BeF₃⁻ bound samples. The 2D [¹⁵N,¹H]-HSQC spectrum of CckA^Rec shows well-dispersed signals indicative of a well-folded protein both for the native and the activated sample (Supplementary Fig. 4). No peak broadening was observed upon activation, arguing for the absence of oligomerization (Supplementary Table 1). Backbone chemical shifts of Cα- and Cβ-atoms were assigned based on standard triple resonance experiments for nearly all residues with the exception of residues forming the acidic pocket (E577, D578, E579) and the phospho-acceptor D623 and the following residues 624 to 628 of the β3–α4 loop. These unassigned residues most likely experience intermediate exchange leading to significant line broadening and thus are not visible in 2D [¹⁵N,¹H]-HSQC experiments.

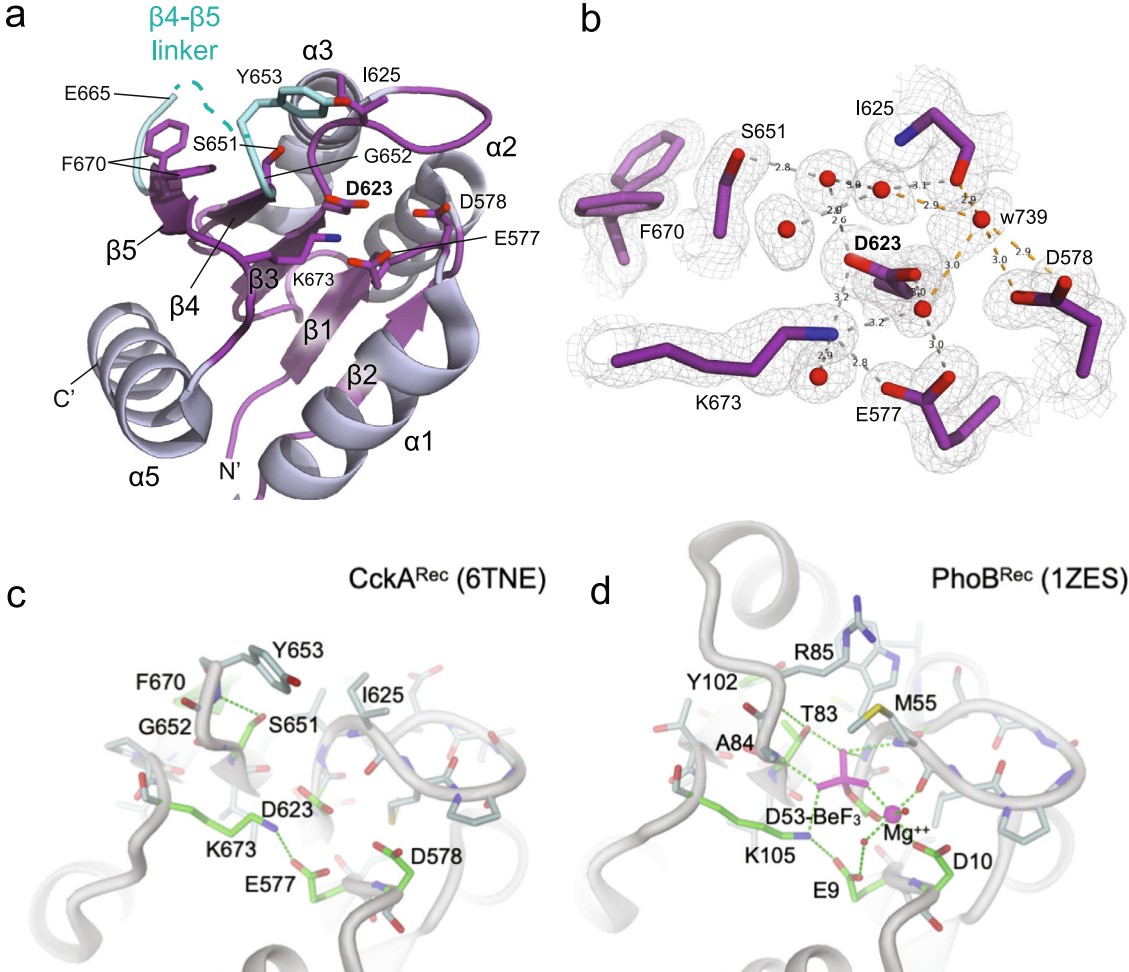

**Fig. 3 | Crystal structure of CckA^Rec. a** Cartoon representation with β-strands colored in purple and α-helices in light-blue. CckA^Rec exhibits the canonical (ßα)₅ fold of response regulator receiver domains, but lacks on ordered α4 helix. The corresponding segment joining β4–β5 is indicated in cyan. The phospho-acceptor D623, acidic pocket residues D578, E579, and K673 implicated in phosphoryl stabilization are shown in full. Also shown are S651 and F670 (with two alternative rotamers) implicated in conformational switching in canonical Rec domains. **b** Detailed view of the phospho-acceptor site with 2Fo-Fc omit map contoured at

1.2 σ overlayed. Several water molecules are well resolved (red spheres) including water W739 which is neares to the canonical Mg^++ binding sites. Potential hydrogen bonds (<3.2 Å) are indicated by dashed lines. **c, d** Full representation of active sites of CckA^Rec (**c**) and activated PhoB (**d**) with selected H-bond and coordination interactions indicated by green, stippled lines. Note that despite the absence of Mg^++ and BeF₃^-, the active site of CckA^Rec is in the activated conformation. See also Supplementary Fig. 3a, b.

Chemical shifts correlate strongly with local structure, of particular value are Cα and Cβ chemical shifts that depend on the protein secondary structure. The NMR-based secondary structure assignments for native CckA^Rec (Fig. 4a, top) are fully consistent with the crystal structure (Fig. 4b), with the exception of the segment connecting β4 with β5, which has significant, but low helical propensity (α4). Upon addition of BeF₃^-, the sequence-specific secondary chemical shifts (Fig. 4a, bottom) remain virtually identical, indicating no change in secondary structure.

Figure 4c shows the chemical shift perturbation (CSB) values obtained from comparing native and activated CckA^Rec spectra. All residues with strong or medium perturbation cluster around the phosphorylation site (Fig. 4d) with residues 648–658 located in β4 and the beginning of β4–β5 linker representing the largest continuous stretch.

To further characterize the backbone dynamics of CckA^Rec, we performed ^15N{^1H}-NOE experiments that report on ps- to ns-timescale motions (Fig. 4e). Strikingly, the residues in helix α4 show faster motions (lower HetNOEs) compared to the rest of the protein, suggesting that this segment is in a rapid disordered loop <-> helix

equilibrium. Thereby, BeF₃^--induced activation shows no significant effect on this mobility (Fig. 4e).

Concluding, the activation does not significantly affect the secondary structure and dynamics of CckA^Rec, but local perturbations are seen at several places around the phosphorylation site, most prominently at the end of β4 and the following β4–β5 linker. The latter segment exhibits moderate helical propensity (corresponding to α4 in canonical Rec domains) and the greatest mobility.

## Intramolecular phosphotransfer within an HHK: DHp/Rec association

The second step in the CckA–ChpT–CtrA/CpdR phosphorelay, after CckA auto-phosphorylation, is the intra-molecular phosphotransfer from H322 to D623 residing on the DHp and Rec domains, respectively (Fig. 1b, c). With the structures of both CckA domains known (5idj[16] and 6tne, this study), we set out to model the functionally competent domain arrangement based on the crystal structure of the complex between ChpT^CA-DHp and CtrA^Rec from *Brucella abortus* (4qpj[14]). Latter complex is thought to represent a phosphotransfer competent DHp/Rec arrangement. The model was generated by individual superposition of the two CckA domains onto the respective domains of the

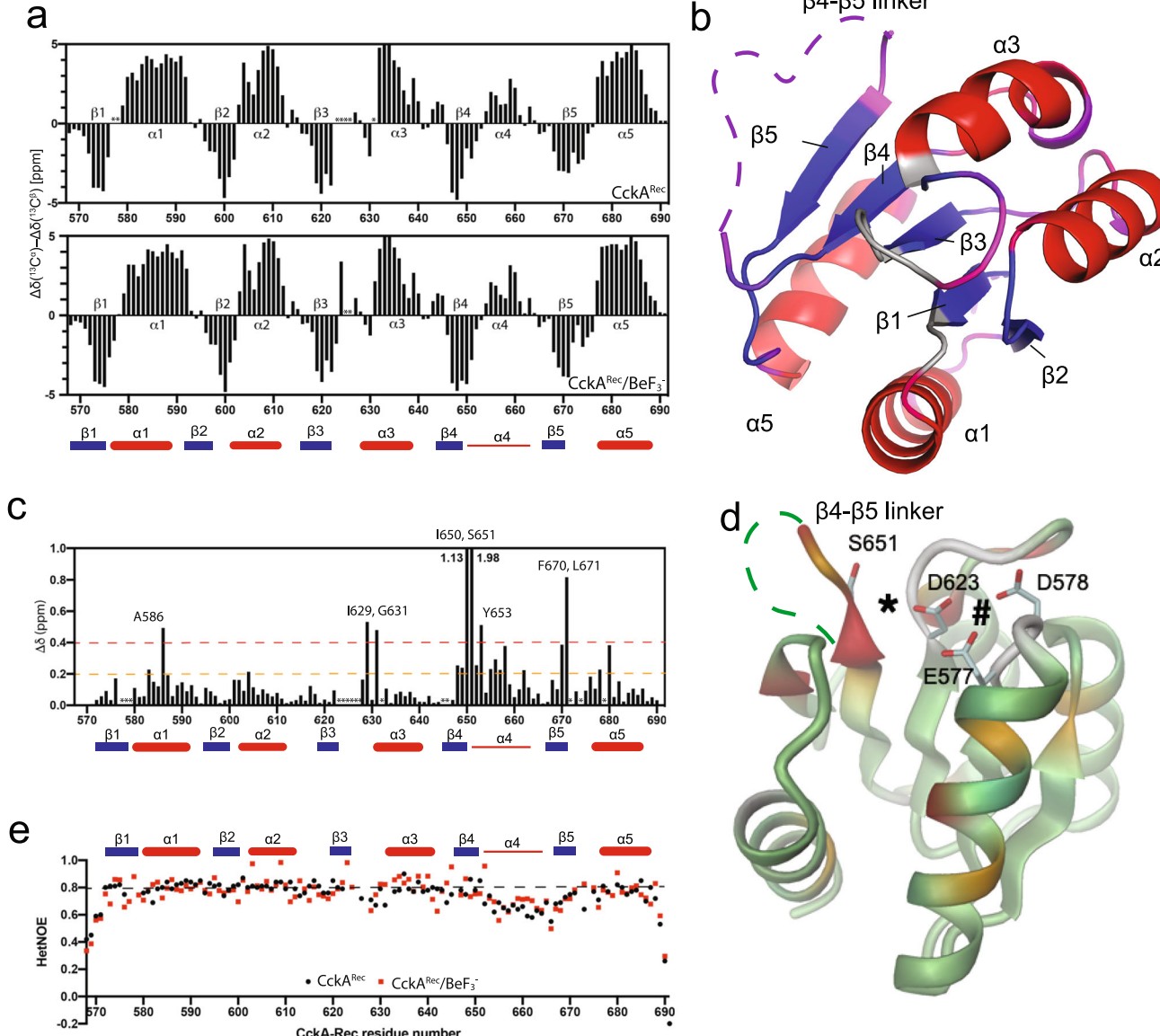

**Fig. 4 | NMR data of apo and BeF₃-activated CckA$^{Rec}$. a** Sequence-specific secondary chemical shifts of apo CckA$^{Rec}$ (top) and BeF₃⁻ activated CckA$^{Rec}$ (bottom) relative to the random coil values of Kjaergaard et al. Unassigned residues are marked with an asterisk. Secondary structure elements as defined by the crystal structure are indicated by thick bars, the low propensity α4 helix as defined by NMR by a thin bar. **b** Structure of CckA$^{Rec}$ with the positions of secondary structure elements identified by NMR spectroscopy indicated by red (α-helices) and blue (β-strands). The β4–β5 loop and other parts with weak positive α-helical propensity are plotted in pink onto the structure. Unassigned residues are shown in gray. **c** Chemical shift perturbation of CckA$^{Rec}$ upon BeF₃⁻ binding. Two significance levels of the chemical shift differences are indicated by the orange and red lines. Not assigned residues are marked with an asterisk. **d** Chemical shift perturbation values of panel c mapped onto CckA$^{Rec}$ structure (0.2 – 0.4 ppm, orange; > 0.4 ppm, red; unassigned regions, gray). Residues of the acidic pocket are shown in full. #: Mg⁺⁺ binding site, *: BeF₃⁻ binding site. **e** Heteronuclear NOEs of apo CckA$^{Rec}$ (black) and activated CckA$^{Rec}$ (red).

experimental complex structure using only interface elements (see Methods) without any side-chain adjustments.

Figure 5 shows the side-by-side comparison of the modeled and the template complex. Even without refinement, the CckA phosphotransfer model (Fig. 5a) shows no clashes between main-chain atoms (including Cβ atoms) of the two domains but favorable contacts between residues that are homologous to interacting residues in the experimental structure. Therefore, it can be inferred that, despite low sequence identity, the domain interactions in the two complexes are analogous involving residues of the lower half (towards the helix connector) of the α1′/α2′ DHp bundle and the Rec α1 helix (Fig. 5a). Strikingly, only one contact is found strictly conserved when comparing the two interfaces. In CckA, this is formed between A330 and A581 (corresponding to A30 and A11 in ChpT/CtrA) with the short side-

chains allowing close contact of the interacting α1 helices (Fig. 5). Other potential contacts involve the conservatively replaced N326/E579 pair (S26/D9 in ChpT/CtrA) and the hydrophobic L333/V585,L589 cluster homologous to the partly polar N33/S15,M19 cluster in ChpT/CtrA. In CckA, an acidic cluster formed by Q337, E350, and E354 may engage in salt-bridges with R588 upon appropriate adjustment of side-chain conformations. In ChpT/CtrA, this is replaced by the apolar L37/L18 contact.

**Promiscuous interactions in the CckA–ChpT– CtrA/CpdR phosphorelay**

The CckA multistep phosphorelay catalyzes transfer of the phosphoryl group between alternating DHp and Rec domains (Fig. 1c), where the domains positioned centrally in the relay engage sequentially with an

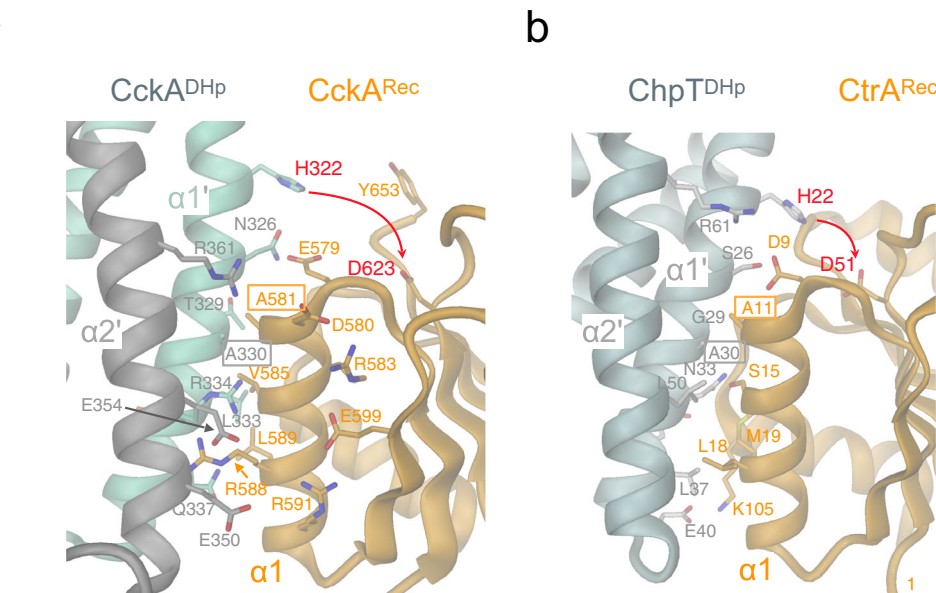

**Fig. 5 | Phosphotransfer competent DHp/Rec association in HHKs.** Cartoon representation (DHp, gray/aquamarine; Rec, orange) with interface residues shown in full. Active histidine and aspartate residues participating in the phosphotransfer are linked by a red arrow. Alanine interface residues, which are conserved between the depicted complexes, have boxed labels. **a** Model of CckA$^{DHp}$/CckA$^{Rec}$ based on the structure shown in panel **b**. The two DHp helices interacting with Rec (α1', α2') are distinguised by color, since they belong to different subunits of the homodimer. Also shown residues of the intramolecular D580–R593–E599–R591 network of CckA$^{Rec}$. **b** Structure of ChpT$^{DHp}$/CtrA$^{Rec}$ (4QPJ). Note, that α1' and α2' of DHp belong to the same subunit, in contrast to the situation in CckA.

acceptor and a donor partner of the same type (e.g., CckA$^{Rec}$ with the DHp domains of CckA and ChpT). The structures of the first three domains of the phosphorelay are now known and the structures of CtrA$^{Rec}$ and CpdR can be determined by homology modeling based on close relatives. Figure 6 shows all structures of the phosphorelay in "open-book" representation to allow easy comparison of the interfaces. Note that the two helices of CckA$^{DHp}$ that are engaged in the interaction belong to separate subunits of the dimeric 4-helix bundle. This is due to the opposite handedness of the helix connector compared to ChpT$^{DHp\,16,23}$. Nevertheless, the helix pairs of CckA and ChpT (Fig. 6) superimpose rather well with an rmsd of 1.3 Å (43 Cα positions).

The DHp/Rec interface of CckA, as derived in the preceding chapter (Fig. 5a), can be divided into four contact patches with the constituting residues color-coded in Fig. 6. From the top, residues N326/E579 (blue) form a polar interaction, A330/A581 (salmon) allow the close approach of the interacting helices, residues Q337, E350, E354/R588 that may form favorable ionic interactions (yellow), and the apolar L333/V585, L589 contact (green). A contact matrix of these CckA domain interactions, and for the other phosphorelay contacts as well. The alanine–alanine contact (salmon), also observed in the *Brucella* ChpT/CtrA complex, is conserved throughout the relay (except for CpdR, where it is replaced by a serine) and appears crucial for the close contact between the interacting helices. The blue intramolecular CckA contact residues are replaced in the other partners but have kept their polar/acidic property. Two of the three acidic CckA$^{DHp}$ residues potentially interacting with R588 of CckA$^{Rec}$ (yellow) are replaced by apolar residues in ChpT$^{DHp}$, thus extending the apolar green patch. This could be important for ChpT$^{DHp}$ to be able to interact with both the largely apolar patch on CtrA$^{Rec}$ and the mixed basic/apolar patches on CckA$^{Rec}$ and CpdR. To summarize, the patches proposed to mediate the intramolecular DHp/Rec contact in CckA, appear involved in all contacts of the phosphorelay. However, in most cases, the interfacial residues are not strictly conserved but conservatively replaced, although there are also a few more drastic substitutions from ionic to apolar residues.

## Two classes of Rec domains in multi-step phosphorelays: Rec$_{inter}$ and Rec$_{term}$

Multi-step phosphorelays (Fig. 1b) involve at least two types of Rec domains with distinct functional roles: (1) intermediary Rec$_{inter}$ domains, which pass the phosphoryl group from an acceptor to a donor, and (2) terminal Rec$_{term}$ domains at the end of the relay. C-terminal Rec domains of HHKs shuttle the phosphoryl group between DHp domains and can, therefore, be classified as bona-fide Rec$_{inter}$ domains. In contrast, Rec$_{term}$ domains are typically part of response regulators and allosterically control the activity of a linked effector domain. How the functional differences between the two types of Rec domains are reflected in their sequences and structures is analyzed below. For a comparative analysis, Hidden Markov Models (HMM) based on the sequences homologous to CckA$^{Rec}$ and CtrA$^{Rec}$ were created to generate corresponding sequence libraries with sufficient variation to reveal class-specific differences (Fig. 7).

All sequences obey the Rec fold as indicated by the overall conservation (gray background in Fig. 7) of crucial Pro, Gly, or hydrophobic residues (CckA: L574, L590, Y595, P627, G631, A654) and show conservation of the phosphorylatable aspartate (D623) and of the residues involved in phosphoryl-stabilization (K673) or magnesium-binding (E577, D578). For Rec$_{term}$ domains, structural S/T–Y/F coupling has been described as an allosteric mechanism for activation, whereby, upon phosphorylation, the S/T residue is pulled toward the phosphoryl group leading to a change in the Y/F rotamer[17]. In all sequences, an S or T residue (S651 in CckA) is conserved at the end of β4 consistent with its role in phosphoryl-stabilization (see Supplementary Fig. 3b). However, while a Y residue is observed on β5 in all CtrA$^{Rec}$ orthologs, there is variability at this position (F670) in CckA$^{Rec}$ orthologs, which would be consistent with the loss of allosteric signaling towards the α4-β5-α5 (4-5-5) face in latter group. Finally, the conserved D/E in CckA position 579 seems to contribute to DHp recognition (Figs. 5 and 6) and the aspartate preceding β3 (D618 in CckA) is probably conserved for structural reasons, as its side-chain is engaged in H-bonding with main-chain amide 572. In summary, and not surprisingly, the overall sequence comparison confirms conservation of the Rec fold and the

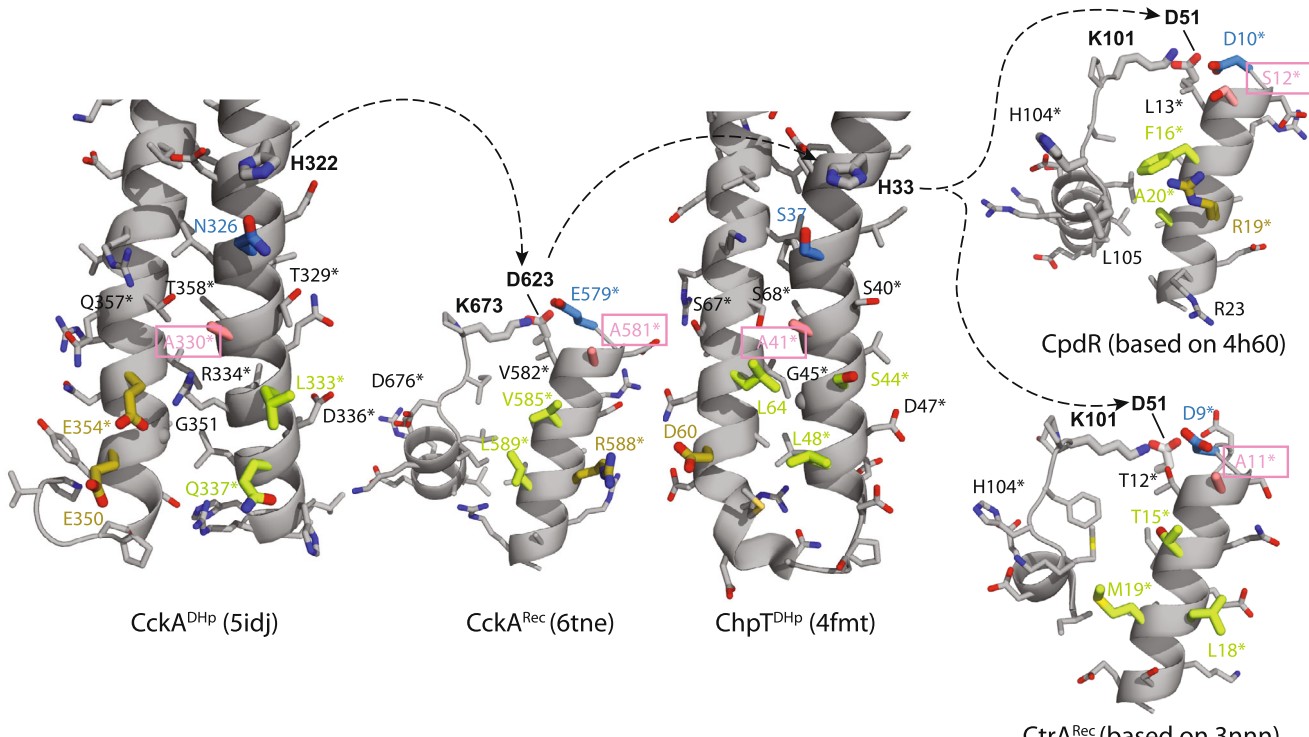

**Fig. 6 | Openbook representation of protein interfaces of the CckA–ChpT–CtrA/CpdR phosphorelay of *C. crescentus*.** On the left, the open book representation of the CckA DHp/Rec model (Fig. 5a) has interacting residues high-lighted by thick bonds and their carbon atoms colored according to individual subcontacts (blue, salmon, green, yellow). The (His-Asp-His-Asp) phosphoryl-transfer path between alternating DHp and Rec domains is indicated by broken arrows. Labels marked by * indicate positions that have been prediced to interact based on co-variation by Skerker et al.[29]. Note, that the two DHp domain of CckA and ChpT superimpose with a rmsd of 1.3 Å (43 Cα positions).

presence of a well-conserved phosphorylation site for both Rec classes.

Comparison of the CckA[Rec] and CtrA[Rec] homologs is most relevant to reveal class-specific positions (Fig. 7). Most of the differentially conserved residues play a structural role (CckA[Rec]: G594, A600, A606, V624, M626, P632, P643, F649, V684; CtrA[Rec]: L54, P74, L78, G94), demonstrating evolution of Rec into two distinct folds that are, however, still closely related (rmsd = 1.5 Å/99 Cα positions and 25% sequence identity between CckA[Rec] and CtrA[Rec] of *C. crescentus*).

More interesting are the positions that may indicate functional specialisation. In CckA[Rec], these are R583 of α1, which is predicted to mediate contact with DHp partners, and the segment 649–655 encompassing part of β4 and the start of the β4–β5 linker showing a FxSGY motif. The segment comprises the phosphoryl-coordinating S651 (S/T position) and Y653, which is exposed and, therefore, probably conserved for functional and not structural reasons. Intriguingly, for the Rec-only proteins Sma0014 and SdrG a similar motif has been identified previously[24,25], see Supplementary Fig. 5b. The motif has been dubbed "FATGUY"[25], a nomenclature we adopt below. This prompted us to generate HMM profiles also for the Rec-only proteins CpdR of the CckA phosphorelay and MrrA, which has recently been shown to constitute a central phosphorylation hub, as it mediates phosphotransfer between various HKs in *C. crescentus*[8]. Intriguingly, also the logos of these proteins (Supplementary Fig. 5a) reveal a FATGUY motif. An HMM profile was also generated for the C-terminal Rec domain of the HHK ShkA, another well-studied HHK in *C. crescentus*[19,26]. This profile clearly lacks the FATGUY motif.

In contrast to CckA[Rec], the CtrA[Rec] logo clearly shows the hallmarks of OmpR-like Rec domains with conserved ionic residues on β5 and α5 that are known to mediate Rec dimerization upon activation[22]. For CtrA[Rec], the predicted inter-subunit salt bridges would be D96–R118, D97–R111 as indicated in Fig. 7. The remaining residues

conserved specifically in CtrA[Rec] sequences are R67 and R117, whose side-chains are H-bonded to main-chain carbonyls of the same or adjacent subunit, respectively, and E107, which forms an intra-molecular salt-bridge with the aforementioned R111.

The HMM-derived logos (Supplementary Fig. 5a) allow classification of individual Rec sequences as shown in Supplementary Fig. 5b. The profiles show good orthogonal discrimination between the founder sequences (Supplementary Fig. 5a, top half), with the exception of some overlap between CckA[Rec] and CpdR. The lower half of Supplementary Fig. 5b shows the classification of some selected, structurally well-characterized Rec proteins. Sma0114[24,27] and SdrG[25] clearly fit the MrrA profile, whereas all HHKs with known C-terminal Rec structure match the ShkA[Rec] profile and are, therefore, not representatives for the CckA[Rec] group.

The conservation properties have been mapped onto the Rec surfaces to demonstrate the similarities and differences of the two classes (Fig. 8). Both classes show a canonical phosphorylation site including an acidic pocket for magnesium binding. Otherwise, surface conservation is completely different between the two classes (indicated in orange and green in Fig. 8). In CckA[Rec], a conserved surface patch is formed by residues of the FATGUY motif from β4/β4-β5 and residues V624, P632. Latter residues, however, may be covered by the flexible β4 - β5 linker. Furthermore, there is residue R583, which is part of an intramolecular salt-bridge in CckA[Rec] and R588, which is most likely involved in DHp recognition (Fig. 5a). In CtrA[Rec], β5 and α5 residues form an extended conserved patch that is involved in dimerization upon activation (see also Supplementary Fig. 3c).

## Co-variation between DHp and Rec residues in HHKs

Despite playing a central role in phosphotransfer, the surface of Rec α1, which engages with cognate DHp domains, is remarkably variable amongst homologs (Fig. 7). This has been attributed to the need to

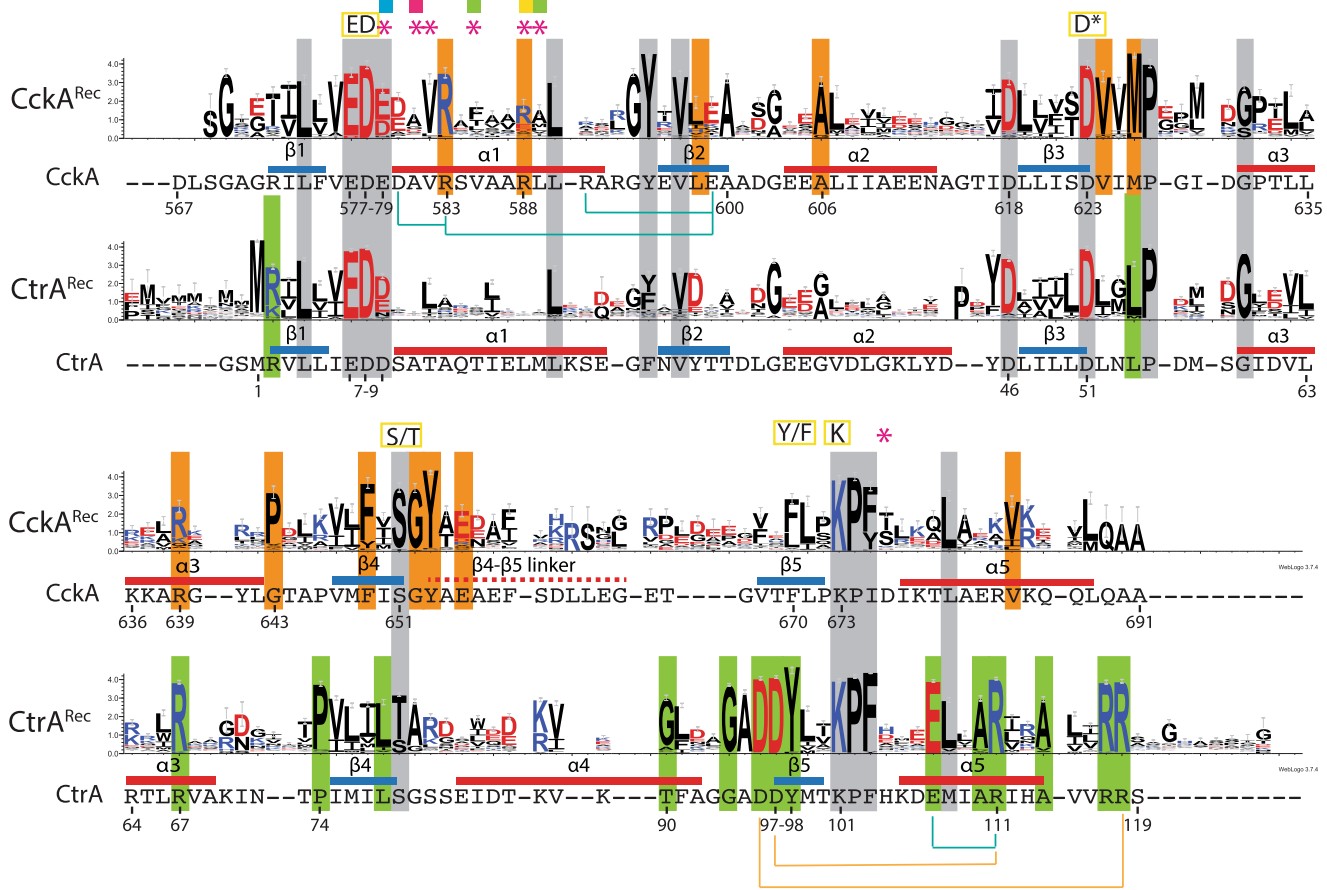

**Fig. 7 | Comparison of CckA^Rec and CtrA^Rec sequence logos.** Overall conserved residues are marked with gray background, whereas specifically conserved CckA^Rec or CtrA^Rec residues are marked with orange and green background, respectively. The relevant part of the CckA and CtrA sequences of *C. crescentus* are reproduced below the logos, with intra-domain salt-bridges indicated by light-blue lines. Also indicated are inter-domain salt-bridges (orange lines; see Supplementary Fig. 3c)

found in activated, dimeric CtrA^Rec structures. The quintet of functionally relevant residues (ED, D*, S/T, Y/F, K), residues co-evolving with cognate DHp domains[26](magenta asterisks), and residues of the sub-contact patches defined in Fig. 6 are marked on the top. The CckA^Rec and CtrA^Rec logos are based on the alignment of 132 and 571 sequences, respectively.

prevent cross-talk between the individual phosphorelays in a given organism[28]. Apparently, cognate contacts can diversify in the course of evolution through co-variation of interacting residues. For two-component systems, this has been demonstrated impressively by Laub and coworkers in their pioneering work more than 10 years ago[29] in which they selected cognate HK and RR pairs based on experimental data or genetic organization (synteny) to study DHp/Rec covariation. For the study of DHp/Rec recognition in HHKs, the cognate pairs are obviously already defined, because they are part of the same polypeptide chain. Since, furthermore, the sequence data base has increased considerably over the last decade, a study on co-variation in HHKs seemed promising.

The EVfold analysis of more than 8000 HHK DHp-CA-Rec sequences (Supplementary Fig. 6) shows, as expected, strong co-variation of residues forming intra-domain contacts, but also between DHp-α1',α2' and Rec-α1 residues. A zoom into the EVfold matrix (Fig. 9a) shows that the hits correlate very well with the DHp/Rec contacts as calculated from the CckA phosphotransfer model (Fig. 5a). Noteworthy, four co-varying pairs (labeled in bold red in Fig. 9a) can be predicted to form salt-bridges in HHKs as deduced from their substitution matrices shown in Fig. 9b. In all four cases, salt-bridges are also observed in the charge reversed configuration (e.g. R/E or E/R are observed about equally often for the pair 341/592), providing further evidence for interaction between these residues in HHKs. Note that, in CckA, only one of these pairs, E350/R588, constitutes an ion pair. To summarize, the straightforward HHK co-variation analysis gave strong,

independent evidence for the validity of the proposed phospho-transfer competent DHp/Rec arrangement. Further studies may exploit the rich information in the substitution matrices of the co-varying residues to understand better how the variation of interface residue types allows specific recognition between cognate domains while preventing cross-talk between phosphorelays.

## Discussion

### Degenerated CckA^Rec fold

Rec domains of RRs constitute the terminal phosphoryl-acceptors of TCSs. Such Rec_term domains have been studied intensively over several decades and hundreds of structures are known, both in the native and the activated form[1]. Typically, upon activation, a conserved S/T residue is pulled towards the phospho-aspartate accompanied by allosteric changes involving the β4–α4 loop and the α4 helix. In addition, a conserved Tyr/Phe rotates inwards to stabilize the active conformation[17,30]. Depending on the particular RR, these changes can then induce a plethora of distinct effects as there is dimerization (OmpR), oligomerization (NtrC), change in dimeric structure (DgcR), relief of effector domain obstruction (CheB), or change in the binding site for a downstream partner (CheY).

Rec domains of HHKs are intermediaries in phosphorelays in that they shuttle the phosphoryl group between DHp or Hpt partners. As an Rec_inter prototype, we have chosen the C-terminal Rec domain of the well-studied HHK CckA from *C. crescentus*[8,16]. CckA^Rec obeys the canonical Rec fold, but with a comparatively low helix propensity and

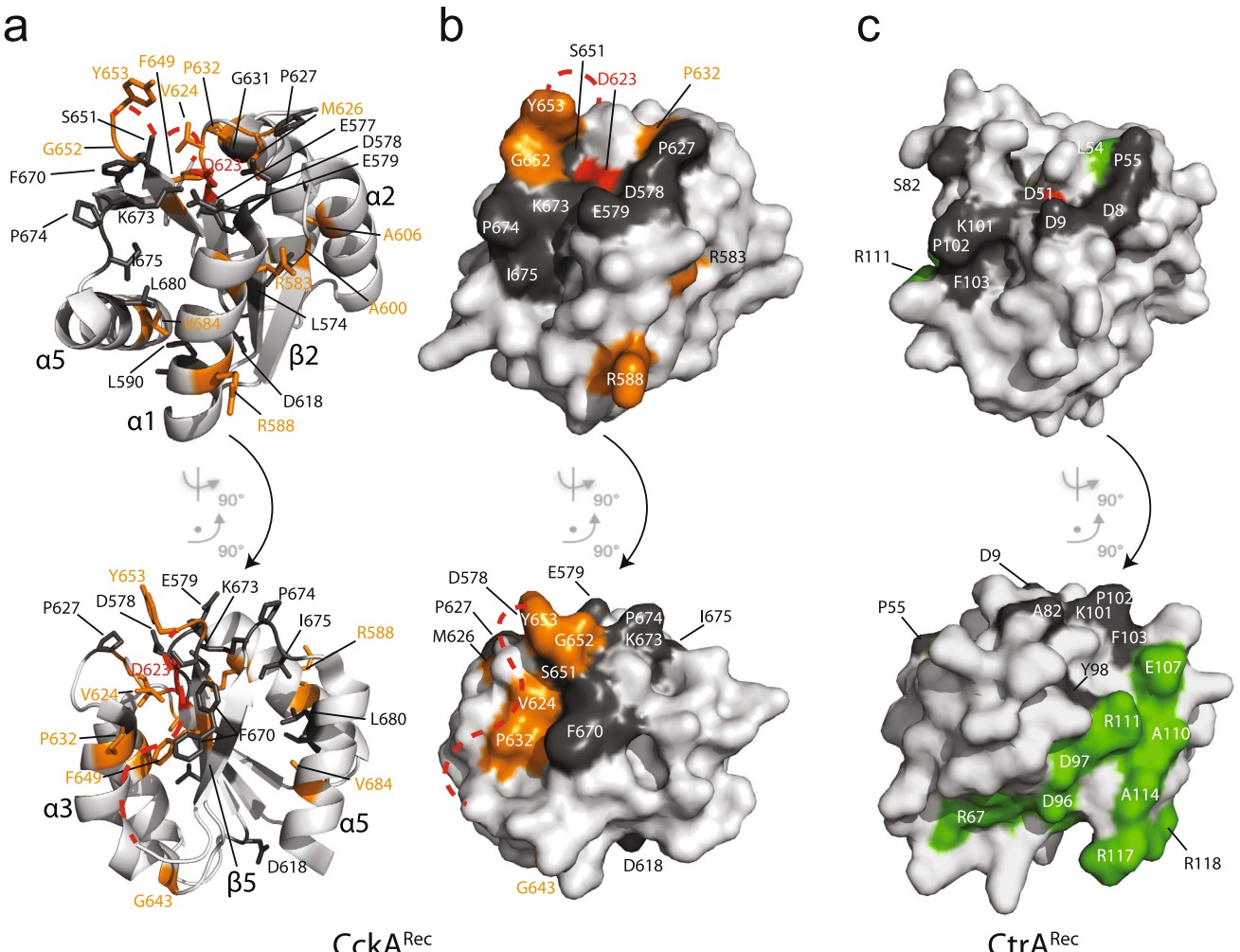

**Fig. 8 | Residue conservation in CckA^Rec and CtrA^Rec mapped onto structures. a** CckA^Rec in cartoon representation with conserved residues in full. **b,c** Surface representation of CckA^Rec and CtrA^Rec with conserved residues colored according to Fig. 7 and the active aspartate in red. The disorered β4–β5 linker of CckA^Rec is indicated by the red dashed line. The top row shows the view onto the α1 face, which would interact with DHp, the bottom row shows the view onto the 4-5-5 face, which is involved in dimerization in CtrA^Rec homologs.

missing electron density for the β4–β5 linker as derived from NMR and X-ray analysis. In canonical Rec domains, this segment is typically folded to a well-defined helix α4. However, a recent comprehensive NMR and modeling study on NtrC[31] suggests that only in the activated state helix α4 is well defined and in fact of maximal length. In contrast, the secondary structure of CckA^Rec remained unaltered upon activation by BeF$_3^-$ suggesting that the β4–β5 linker is not subject to allosteric alteration.

Missing α-helices have been reported for other Rec domains (Fig. 2), but to our knowledge only for Rec-only proteins (e.g. Sma0114 and SdrG, see further below) or accessory (phosphorylation-incompetent) Rec-like domains of HHKs (RcsC[20] and ShkA[19]). Thus, the Rec-fold appears to be a stable framework tolerating the loss of helices and such variation may indicate neutral evolution or functional specialization. For CckA, the latter reason can probably be excluded, since the β4–β5 linker (with the exception of the first few residues) is highly variable and not in a position to interact with phosphotransfer partners (see the SpoOB/SpoOF and ChpT/CtrA^Rec complex structures[14,15]).

### Lack of allosteric response to modification

Unusual for Rec domains, unmodified CckA^Rec shows the active site in the activated conformation with the β3–α3 loop closed down and the conserved S651 at the end of β4 in a position to coordinate a phosphoryl group bound to the active aspartate (Fig. 3c, Supplementary

Fig. 3). Although a preformed active site has also been reported for unmodified PhoP[32], However, its active conformation was probably induced by the high concentration used for crystallization causing formation of the canonical dimer via isologous interactions of the 4-5-5 faces and, thus, represents an example of allosteric backward signaling. Such an artifact can be excluded for CckA^Rec since it is monomeric in the crystal lattice as it is in solution. With a fully preformed active site, no significant structural changes would be expected upon phosphorylation or BeF$_3^-$ modification, consistent with the NMR chemical shift perturbation analysis (Fig. 4). In summary, unlike the well-known phosphorylation-induced allosteric response at the 4-5-5 face in Rec$_{term}$ domains[17], there is no evidence that changes of similar magnitude take place in CckA^Rec and the same can be expected for its orthologs. Since mainly α1 is involved in the interaction with the phosphorelay partners (Figs. 5, 9), changes at the distant 4-5-5 face would probably have no effect on phosphotransfer and would therefore be dispensable.

### Phosphotransfer complex model and promiscuity of HHK Recs

The intramolecular CckA DHp/Rec association has been modeled based on a known phosphotransfer complex (ChpT/CtrA^Rec complex from B. abortus[14]. It has to be noted that, in the modeled complex, the His-Asp distance is too long for associative phosphotransfer (10.4 Å compared to 5.9 Å in ChpT/CtrA^Rec). The difference in the histidine

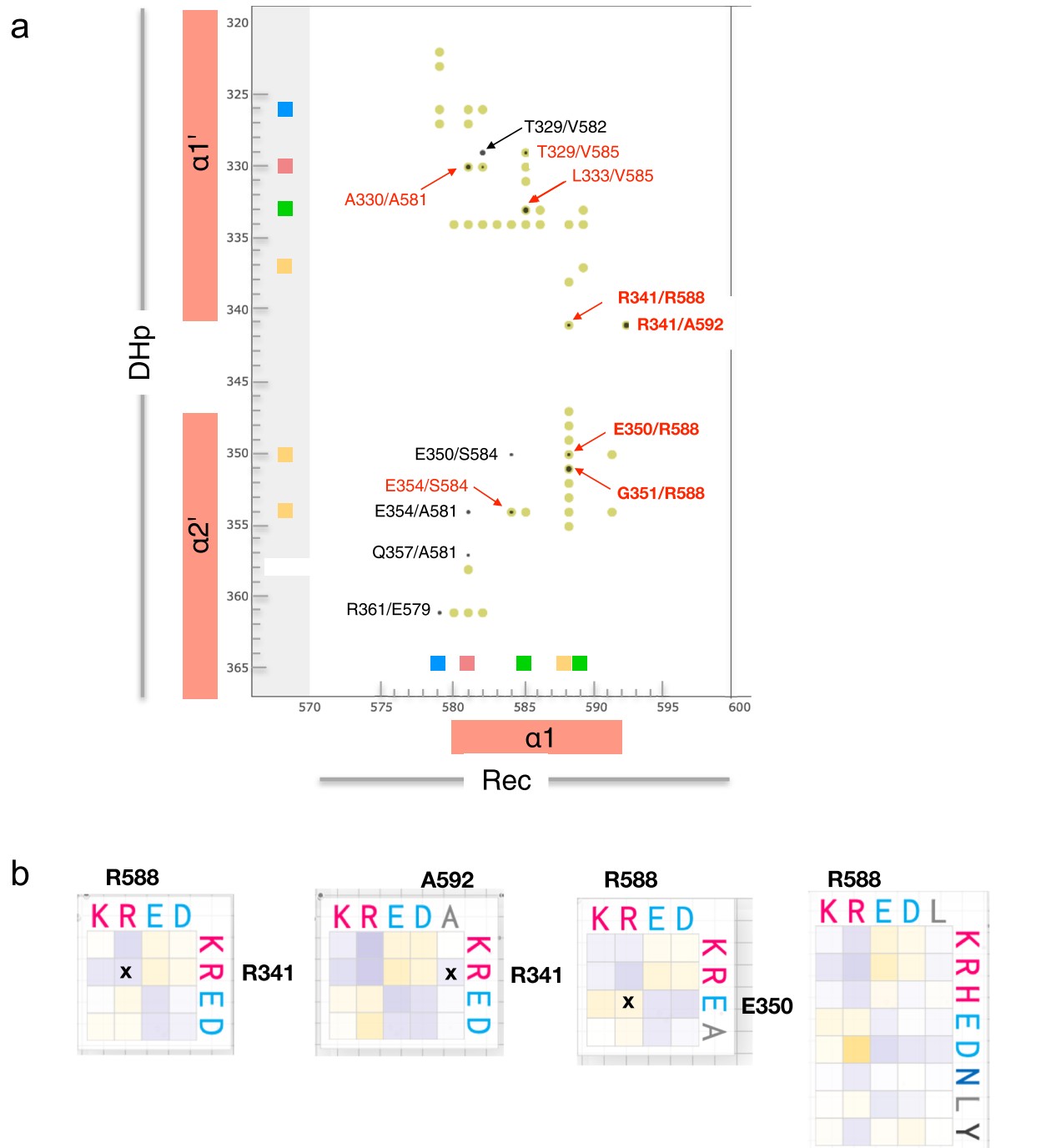

**Fig. 9 | DHp/Rec contacts as revealed by co-variation analysis of HHK sequences. a** Co-variation scores above 3.0 between DHp (vertical axis) and part of Rec (horizontal axis) residues are shown as filled circles, with size proportional to signal strength. Residues of the sub-contact patches defined in Fig. 6 are indicated by correspondingly colored squares on the axes. Distances <6 Å in the CckA$^{DHp}$/CckA$^{Rec}$ model (Fig. 5a) are indicated by filled yellow-green circles. The full co-variation matrix is given in Supplementary Fig. 6. Co-variation peaks that correspond to close distances are labeled in black (<8 Å) or red (<6 Å). Contacts that constitute potential ion-pairs are labeled in bold with further information given in panel **b. b** Substitution matrices for selected DHp/Rec residue pairs. Note that, e.g., R341/A592 do not form a contact in the model (due to the short A592 side-chain), but position 592 is often occupied by a D or E.

position appears to be due to a distinct orientation of the N-terminal end of CckA α1′ preceding a smooth bend around residue 328 (in ChpT, there is a sharp kink due to the presence of a proline in position 27 corresponding to CckA residue 327). It is well conceivable that a tertiary change in the DHp domain of CckA, i.e. change in α1′ bending, can be induced by Rec/CA interactions. Such interactions have not been considered here and are not evident in the covariation analysis (Supplementary Fig. 6), but could move the histidine into the position required for phosphotransfer. In the ChpT/CtrA$^{Rec}$ complex, residues of the Rec β4–α4 loop interact with the CA-domain[14]. In CckA, such interactions could involve the Y of the FATGUY motif and subsequent residues from the Rec β4–β5 linker and may contribute to affinity, but

also to α1' bending for efficient phosphotransfer. Alternatively, the relative orientation of the DHp and Rec domains in CckA may not be exactly the same as in the ChpT/CtrA(Rec) template complex. This notion is supported by the observation that the relative domain orientation within the experimental SpoOF/B and ChpT/CtrA complexes differs by about 30° although the same interface is used (Supplementary Fig. 7).

Multi-step phosphorelays involve at least two domains that act as intermediaries (e.g. the two central domains of a His-Asp-His-Asp phosphorelay) and interact successively with two partners of the same fold. We have shown that the involved interfaces show surprisingly little conservation (Fig. 6) suggesting that the partners are evolutionary not particularly close. Apparently, based on the same frameworks, there are various solutions to form catalytically competent domain associations by residue-type variation. This has been demonstrated already very convincingly by co-variation analysis of cognate HK/RR pairs[29] and has been associated with the need to insulate the various TCSs of a given organism against mutual interference. Our HHK analysis (Fig. 9a) shows very clear intra-domain covariation, probably due to the large number of sequences considered, and is broadly consistent with the previous results on canonical HK/RR pairs[29] and the phosphotransfer model (Fig. 5a). Individual substitution matrices (Fig. 9b) demonstrate the extensive repertoire of residue combinations for selected pairs, including charge reversal of salt bridges.

Lower specificity of the intramolecular recognition in HHKs compared to cognate HK/RR pairs has been reported and attributed to the relaxed constraints for phosphotransfer due to the high local concentration of the covalently attached Rec domain[32,33]. In both studies, covariation between HHK DHp and Rec residues was found to be much lower compared to canonical (non-tethered) pairs. Here, by being able to analyze a vast number of sequences, we have found very significant covariation between the tethered DHp and Rec domains of HHKs (Fig. 9). It would be interesting to see, whether covariation between cognate DHp/Rec from separate proteins turns out to be even more pronounced, when using a similarly large database. Also, we note that most HHK sequences available at the time of the earlier studies[32,33] covered a wide phylogenetic range, which raises the question whether all those proteins use equivalent contact points for the intra-molecular recognition, a pre-requisite for a covariation analysis, whereas in our study we used a more closely related group of sequences. In any case, it should be considered that an HHK Rec domain not only has to recognize its covalently attached DHp domain, but also its specific downstream HPt partner, with latter interaction subject to the same evolutionary constraints as in canonical HK/RR systems.

## Rec classification

Most Rec domains can be identified easily by the match of their sequences with the pfam motif pf00072, which covers the active site and residues crucial for the $(\beta\alpha)_5$ sandwich fold. A very useful subclassification of Rec domains has been performed on the basis of their domain context by Galperin (https://www.ncbi.nlm.nih.gov/Complete_Genomes/RRcensus.html[2]. However, it has not yet been investigated whether there are intrinsic Rec features that correlate with this subclassification. In the Conserved Domain Database of the NCBI, a very finely graded Rec sub-classification of the Rec entry (cd00156) with 82 "models" based on sequence similarity can be found (https://www.ncbi.nlm.nih.gov/Structure/cdd/cdd.shtml)[34]. However, many of the subgroups consist of only a few closely related sequences.

For our comparative Rec sequence analysis (Fig. 7), the chosen threshold for sequence selection resulted in a moderate mean pairwise identity of about 45% within the subgroups. Such sequence variability made it possible to uncover subgroup specific motifs and to derive well-discriminating HMM profiles. Indeed, searching with the CckA^Rec profile against the Reference Proteome database (as implemented in HMMER) using an E-threshold of 1e-25 retrieved HHKs and Rec-only proteins exclusively. Conversely, when searching with the CtrA^Rec profile, as expected RRs were found, but also HKs with N-terminal, but not C-terminal, Rec domains were found.

As expected, the CtrA^Rec profile is specific only for the OmpR-Rec class and not for Rec_term domains in general. While probably all Rec_term domains have an inbuild allosteric mechanism to control their output domain function, the details appear to be diverse. Since part of the CtrA^Rec profile comprises residues that allow OmpR-Rec domains to homo-dimerize via the 4-5-5 face, Rec_term domains with other allosteric mechanisms (NtrC, CheY, etc.) will not be recognized by the CtrA^Rec profile.

Similarly, not all Rec_inter domains conform to the CckA^Rec profile. In fact, the C-terminal Rec domains of most *C. crescentus* HHKs show similarity to the corresponding domain of ShkA and not of CckA. It would be interesting to learn more about the phosphotransfer mechanism of ShkA^Rec domains. In particular, their resemblance to CtrA^Rec (i.e. OmpR-type Rec domains) is intriguing. Thus, as with the Rec_term domains, the Rec domains that act as intermediaries in phosphotransfer are not a monophyletic group.

## Rec-only proteins and FATGUY motif

Rec-only proteins form a large group with diverse functions. A prominent group is the chemotactic CheY proteins that regulate bacterial motility. CheY phosphorylation affects the affinity of e.g. FliM motor peptides due to an allosteric change of the binding site at the 4-5-5 face accompanied by Y-T coupling similar as in transcriptional RRs. The Rec-only protein CpdR of the branched CckA phosphorelay is required to enable ClpXP-mediated CtrA degradation[11]. Thereby, CpdR seems not to act as a classical adaptor for CtrA degradation (e.g. SspB for SspA tagged proteins[35]), but rather prime the proteolytic complex for this activity. Since CpdR phosphorylation interferes with the priming function, both branches of the CckA phosphorelay work together to ensure tight transcriptional regulation (Fig. 1c). It has been shown that CpdR interacts with the N-terminal Zn-finger domain of ClpX[36], but only in its non-phosphorylated state. No details about the binding mode are known and whether phosphorylation interferes directly or allosterically with binding, though residues on α5 seem involved in the interaction arguing for an allosteric effect[36].

The sequence logo of CpdR does not reveal additionally conserved residues that could potentially interact with ClpX (Supplementary Fig. 5a, compare with the CtrA^Rec logo with its conserved residues at the C-terminus mediating homo-dimerization). Either such residues are co-varying with ClpX or residues of the CpdR active site are mediating the contact with ClpX, which would also explain the dependence of binding on the CpdR phosphorylation state. Strikingly, CpdR reveals a FATGUY-like motif at β4 and the following loop as seen in CckA^Rec. Given that both domains have to engage with ChpT for phosphotransfer, it is likely that the motif is related to this specific function, especially since the overall sequence similarity of the two domains is only moderate (35%).

A role of the FATGUY motif in phosphotransfer between Rec and DHp or Hpt domains would be consistent with the occurrence of the motif in other, well-studied proteins that act as Rec-only intermediaries: the phosphorylation hub MrrA[10], SdrG[25,37] from the general stress response and Sma0114[27]. Latter two proteins have been studied by NMR in their native and activated form. For Sma0114, it was initially proposed that the FATGUY motif would indicate an alternative mechanism to Y-T coupling, considering a missing α4 helix and a missing aromatic residue on β5[24]. Such a mechanism, however, was refuted by the subsequent investigation into structure and dynamics of the activated form of Sma0114 that showed only minor changes in the motif[27]. For SdrG, a change in the orientation of T83 and a repacking of a (not conserved) phenylalanine from the β4–β5 linker (F94) with the hydrophobic core upon activation has been reported[25].

However, the considerable variation in the ensemble of the activated structures makes a detailed description of activation-induced changes difficult. We note that all aforementioned NMR structures were modeled with a trans-peptide bond between lysine 102 and the following proline, although a cis-peptide bond between these two conserved residues is a hallmark of the Rec-fold[17] and is required to place the conserved lysine in the active site (Fig. 3c, d). Thus, the proposed role of the lysine to switch from an outward to an inward orientation upon activation in SrdG[10] has to be questioned. SpoOF from *B. subtilis* is another well-studied single-domain Rec$_{inter}$ protein. Although in an initial NMR study[38] significant shifts of helices α1, α3, α4 were reported upon BeF$_3^-$ modification, this was not confirmed later by the crystallographic study of Varughese and coworkers[15]. Surprisingly at that time, the BeF$_3^-$-bound SpoOF structure superimposed closely with native SpoOF (both determined in complex with SpoOB)[39] (Supplementary Fig. 7).

In summary, the significance of the FATGUY is still not resolved. It is present in Rec$_{inter}$ domains (such as CckA$^{Rec}$, MrrA, Sma0114, and SdrG), which most likely do not rely on an allosteric control of the 4-5-5 face for their phosphotransfer function. The same may be true for CpdR, though positioned at the end of the phosphorelay. Sterically, it appears possible that residues from the end of the motif (in particular the tyrosine) interact with the CA domains of their phosphotransfer partners to enhance affinity, but perhaps also to control DHp conformation allosterically.

Altogether, our findings suggest that intermediary Rec domains play a rather passive role in phosphotransfer. Phosphorylation of their preconfigured active site will not induce structural changes such as the inward movement of the S/T residue and concomitant allosteric changes in Rec$_{term}$ domains evolved to control output domain function or partner recognition. Here we have analyzed the Rec$_{inter}$ domain of CckA and its orthologs. Future structure/function studies on other Rec$_{inter}$ families will show whether these findings can be generalized and will allow further sub-classification of Rec domains to allow comprehensive sequence-based assignment of Rec function (Rec$_{inter}$ versus Rec$_{term}$) in multi-step phosphorelays.

## Methods

### Cloning and protein expression

The DNA fragments corresponding to the receiver (Rec) domain (568-691) of wild-type CckA (Uniprot: Q9X688) was amplified first by PCR using the primers pairs, Forward: Q5SDM_CckARec_F; Reverse: Q5SDM_CckARec_R. To add a N-terminal His-tag, a second PCR was applied on the first PCR product by using the primers pairs, Forward: Q5SDM_CckARec_N-histag_F; Reverse: Q5SDM_CckARec_N-histag_R (Supplementary Table 2). The second PCR product corresponding to 6-His_CckA-Rec was treated with DpnI to eliminate template plasmids and cloned using ligation-independent cloning[40] into pET28a vector (Novagen) yielding pET28-CckA-Rec plasmid. After successful cloning, pET28a-CckA-Rec plasmids were extracted from *E. coli* DH5α cells following QIAprep Spin Miniprep Kit protocol (QIAGEN) and used to transform the expression strains *E. coli* BL21(DE3) or similarly efficient Rosetta cells (Novagen). For protein expression, adequate amounts of LB-Kan media were inoculated with 1% pre-culture of transformed cells. Cultures were grown at an incubation temperature of 37°C and induced with 1 mM isopropyl 1-thio-β-D-galactopyranoside (IPTG) upon reaching an OD$_{600}$ of 0.6–0.8. The incubation temperature was reduced to 21°C for overnight protein expression. Cells were harvested by centrifugation at 9000 RCF for 10 min at 4 °C. Pellets were stored at −20 °C or lysed immediately.

### Protein purification

Purification was entirely performed at 4 °C. Pellets were homogenized in lysis buffer containing immobilized metal affinity chromatography (IMAC) loading buffer (500 mM NaCl, 30 mM Tris-HCl, 5 mM MgCl2, 20 mM Imidazole, pH 7.5), complete EDTA-free protease inhibitors (Roche) and bovine pancreas DNase I (Roche). Cell lysis was performed using French-press (3 passes, 10,000 psi). The lysate was ultra-centrifuged at 31,000 RCF for 1 h (Thermo Fisher Scientific centrifuge), to remove cell debris and suspended particles. The clear supernatant was applied on a 5 mL Ni-NTA column (GE Healthcare) pre-equilibrated with IMAC loading buffer. Bound protein was eluted with a linear gradient of IMAC elution buffer (500 mM NaCl, 30 mM Tris-HCl, 5 mM MgCl$_2$, 500 mM Imidazole, pH 7.5) using an ÄKTA Purifier system (GE Healthcare). Fractions containing the desired protein were pooled and concentrated to a volume of 5 mL. The concentrated protein was centrifuged at 16,000 × *g* at 4 °C for 15 min and loaded onto a Super-dex 75 gel filtration column (Amersham Biosciences) equilibrated with 30 mM Tris/HCl pH 7.5, 5 mM MgCl$_2$, 100 mM NaCl, 1 mM DTT. The concentrations of the collected samples were quantified by UV absorption with a NanoDrop 2000 spectrophotometer (Thermo Fisher Scientific) and either used freshly (e.g. for crystallization) or stored at −80 °C.

### Crystallization

Protein solubilised in SEC buffer (see above) was crystallised using the sitting-drop vapor diffusion method. Sets of 3-drop MRC plates were prepared with a Gryphon robot (Art Robbins Instruments). CckA$^{Rec}$ was crystallised at three concentrations 20, 10, and 5 mg/mL, with a crystallization mixture consisting of 0.15 M KSCN, 0.1 M Na Cacod 6.5 pH, 20% v/v PEG 600 (Clear Strategy II-D5) at room temperature.

### Data collection and structure determination

All single crystal X-ray diffraction data sets were collected at PXI beam line of Swiss Light source, Villigen, Switzerland.) at 100 K. Diffraction data sets were processed either with iMOSFLM[41] or XDS[42], and the resulting intensities were scaled using SCALA from CCP4/CCP4i2 suite[43]. The crystal structure of CckA$^{Rec}$ was solved by molecular replacement using native structure of DgcR (PDB code, 6ZXM[3]) as a search model using Molrep[44]. Structure refinement was carried out using REFMAC5 of the CCP4 suit[43] and finally Phenix[45] with anisotropic B-factors and optimisation of the X-ray/stereo-chemistry and ADP weights. Model building was performed using COOT[46] and model validation was carried out with Molprobity[47]. Crystallographic data processing and refinement statistics are provided in Table 1.

### Solution NMR spectroscopy

Protein samples were prepared in 30 mM MES buffer at pH 6.8 with 100 mM NaCl and 5 mM MgCl$_2$ in 5%/95% D$_2$O/H$_2$O with a protein concentration of 0.8 mM. All NMR spectra were recorded at 25 °C on a 700 MHz Bruker Avance spectrometer equipped with a cryogenic triple-resonance probe. For the sequence-specific backbone resonance assignment of [*U*-$^{15}$N,$^{13}$C]-labeled CckA$^{Rec}$ with and without bound BeF$_3^-$, the following NMR experiments were recorded: 2D [$^{15}$N,$^1$H]-HSQC, 3D HNCACB, and 3D CBCA(CO)NH. Chemical shift differences of amide moieties were calculated as $\Delta\Delta(HN) = ((\delta(^1H_{ref}) - \delta(^1H))^2 + ((\delta(^{15}N_{ref}) - \delta(^{15}N))/5)^2)^{1/2}$. For the characterization of backbone dynamics of [*U*-$^{15}$N,$^{13}$C]-labeled CckA$^{Rec}$ with and without bound BeF$_3^-$, $^{15}$N{$^1$H}-NOE experiments were measured. Data were processed using Prosa[48] and analyzed with CARA (Keller R. 2004). The backbone assignment was done manually using CARA. Combined secondary chemical shifts of $^{13}$C$^\alpha$ and $^{13}$C$^\beta$ were calculated relative to the random-coil values of Kjaergaard and Poulson[49]. A 1-2-1 smoothing function was applied to the raw data for the display in Fig. 4a. For the analysis of NMR line widths to infer on the oligomerization state, from the 2D [$^{15}$N,$^1$H]-HSQC spectra of inactive and activated CckA$_{Rec}$ 1D slices of the F1 ($^{15}$N) – dimension were extracted that contained the respective residues to be

analyzed. The 1D amide proton signal was fitted using a gaussian function and the line width at half maximal intensity was calculated using Topspin 4.0.

## SEC-MALS

SEC-MALS measurements for CckA$^{Rec}$ were performed at a sample loading concentration of 13.8 mg/ml; 6.9 mg/ml and 3.5 mg/ml at 25 C in sample buffer using a GE Healthcare Superdex 200 10/300 Increase column on an Agilent 1260 HPLC. Elution was monitored using an Agilent multi-wavelength absorbance detector (data collected at 280 and 254 nm), a Wyatt Heleos II 8+ multi-angle light scattering detector and a Wyatt Optilab rEX differential refractive index detector. The column was equilibrated overnight in the running buffer to obtain stable baseline signals from the detectors before data collection. Inter-detector delay volumes, band broadening corrections, and light-scattering detector normalization were calibrated using an injection of 2 mg/ml BSA solution (Thermo-Pierce) and standard protocols in ASTRA 6. Weight-averaged molar mass (Mw), elution concentration, and mass distributions of the samples were calculated using the ASTRA 6 software (Wyatt Technology).

## Modeling of the phosphotransfer competent CckA DHp/Rec complex

The model of the CckA DHp/Rec complex was generated by (1) superposition of the CckA$^{DHp\_CA}$ monomer structure (5idj) onto the A-chain of ChpT from the ChpT/CtrA$^{Rec}$ structure (4qpj) using only the Cα positions of the C-terminal half of α1' and the N-terminal half of α2' of the DHp domains. These elements are part of the interface in the ChpT/CtrA$^{Rec}$ structure. (2) superposition of CckA$^{Rec}$ (6tne) onto CtrA$^{Rec}$ (C-chain of 4qpj) using only the Cα positions of the Rec α1 helices, since this helix is part of the interface in the experimental complex. (3) reassembly of the individually superimposed domain structures into a single file.

## Bioinformatics

Sequence logos of Rec domains (Supplementary Fig. 5a) were calculated from the alignment of homologous sequences obtained by successive phmmer and hmmsearch calculations (using HMMER from the EBI website https://www.ebi.ac.uk/Tools/hmmer/,44) against the Reference Proteomes data base starting with the respective sequences of the *C. crescentus* homologs using E thresholds between 1E-20 and 1E-25 depending on the group. In the case of CckA$^{Rec}$, only full-length sequences were selected that had the three essential HHK domains (HISKA, HATPase_c, Response_reg) but no additional domains annotated. For CpdR and MrrA, single domain proteins with Response_reg annotation were selected. HMM profile were generated using hmmbuild and were run against selected sequences (Supplementary Fig. 5b) using hmmscan from a local hmmer (version 3.1) installation. Co-variation in HHK sequences was analyzed by the EV fold webservice https://v2.evcouplings.org/[50]. Details are given in the respective figure legends.

## Reporting summary

Further information on research design is available in the Nature Portfolio Reporting Summary linked to this article.

## Data availability

Coordinates and crystal structure factors of CckA$^{Rec}$ have been deposited in the Protein Data Bank (PDB) under the accession number 6TNE. Sequence-specific resonance assignments have been submitted to the Biological Magnetic Resonance Data Bank (BMRB) under accession number 51627 for apo CckA$^{Rec}$ and 51628 for CckA$^{ReC}$/BeF$_3^-$. The structure of DgcR used for molecular replacement is available under PDB accession number 6ZXM.

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

## Acknowledgements
We thank the beamline staff at the Swiss Light Source in Villigen for expert help in data acquisition and T. Sharpe from the Biophysics facility at the Biozentrum Basel for expert biophysical support.

## Author contributions
M.B. performed purifications, crystallization, and biophysical experiments under supervision of B.N.D. B.N.D. and M.B. processed x-ray data, determined structure, built and validated the model. T.S. and M.B. performed bioinformatics analysis. R.B. performed NMR experiments. F.F. performed cloning. R.B. and S.H. analyzed NMR data. B.N.D., M.B., and T.S. wrote the manuscript with input from all authors. B.N.D. and T.S. conceived and directed the project.

## Funding

## Competing interests
The authors declare no competing interests.
