## [Peer Review File · Nature Communications]

REVIEWER COMMENTS

Reviewer #1 (Remarks to the Author):

Brüderlin and collaborators report Rx crystallographic and NMR data studying the intermediary REC domain of the hybrid his-kinase (HHK) CckA from *Caulobacter crescentus*. They also perform clever sequence bioinformatics analyses, extending their observations to understand more general principles that might discriminate between intermediary Asp-containing response regulators (in phosphorelay pathways) and terminal RRs (the ones producing TCS and phosphorelay pathways' output effects).

The manuscript is well written and, supported on a beautiful set of illustrations, the messages are indeed very clear.

The knowledge gap the authors have set out to fulfill is based on the fact that intermediary REC domains have received less attention compared to terminal ones. While the goal is interesting, there is not a clear hypothesis behind the study.

In particular, the authors don't seem to exploit previous work in the field, which I believe is relevant, and that would've helped in defining a clearer set of premises and hypotheses for their own work. They do refer to results reported by the Laub group (Capra et al doi:10.1111/mmi.12064), nicely showing now, that HHKs do reveal amino acid coevolution among their tethered kinase and receiver domains (interestingly contrasting to Capra and coworkers' hypothesis, likely due to the use of smaller sequence datasets than available). However, this is not discussed in the context of the reduced constraints in the specificity of these domains' binding that early reports had shown (such as in the 2013 paper by Townsend et al. doi:10.1073/pnas.1212102110, which is not cited).

Most importantly, the 2021 report by Ortet et al (doi:10.1038/s41598-021-91260-w) has also been overlooked, even though that paper, using similar kinds of co-variance analyses in HHKs, had already shown quite convergent observations and conclusions to the ones the authors are now presenting. For instance, what Brüderlin et al conclude here in lines 384-393, goes very much in the same direction as the Ortet et al paper did, in the sense that inter-domain specificity has been selected for during evolution, also in intra-molecular HHK associations.

Besides the RECinter-DHp/HPt specificity analysis, the effect of phosphorylation in stabilizing/triggering conformational changes of RECinter domains had also been analyzed in a number of papers. I will mention one by Gardino et al in 2003 (not referenced here), where extensive analysis of the *B subtilis* SpoOF (a monodomain RECinter protein, within the phosphorelay pathway that regulates sporulation) was performed by NMR. Gardino et al do identify conformational variations associated to phosphorylation, and also focus on helix alpha4 –which they see by NMR– already pinpointing that it is a less stable helix. I also wonder whether Brüderlin et al

also observe this helix $\alpha 4$ by NMR now (it appears to be the case, according to lines 143-144). Maybe the helix is there in the crystal structure but given its higher flexibility, its electron density is weaker? (see below, but maybe a more complete/accurate model-building job, resulting in lower R factors, could eventually reveal helix $\alpha 4$'s density?).

Overall then, the new insights coming from this manuscript are not that clear to me.

Major comments:

- Refinement at this resolution (which is well within atomic resolution; particularly considering that this crystal most likely diffracted to higher resolution than what the authors decided to collect, according to highest resolution shell statistics), requires special attention to standard protocols :

- o Rwork and Rfree seem quite high (the validation report picks this up immediately of course);

- o B-factors should most probably be refined anisotropically (if a significant drop in Rfree is achieved by doing so);

- o several clashes that are now observed in the validation report, involve water molecules: they could be indications of double/multiple conformations for several side chains (typically observed at atomic resolution);

- o as it is now, the model includes quite low a number of ordered water molecules. Especially considering that the average B factor is quite low. With the figures reported on Table 1, I would expect that for a ~ 120 -residues protein, we would be able to see in the range of >200 water molecules. For this purpose, it would be important to check how the difference Fourier map looks like at this stage of refinement, in terms of unaccounted peaks (i.e. how many peaks does the mFo-DFc map show, e.g. counted at 3.5σ , and what heights do those peaks display?). Such analysis would objectively gauge whether there are significant features that could still be (re)modelled.

None of these protocols, standard for atomic resolution data, are mentioned in the Methods section, so I assume they were not considered. These comments are not mere technicalities, but rather anchored on the important concept that phases improve as the model gets more complete and accurate. Better phases impact on the quality of the whole map, including any regions that might so far be less well defined (the $\beta 4 - \beta 5$ linker? and all others).

- I2 is an infrequently used (body-centered) setting of monoclinic space groups: by definition it possesses a 2-fold (dyad) axis. Please correct (lines 108-111; this misconception is also implicit in line 350, I suspect). Hence, it does seem relevant to analyze whether this dyad generates a dimeric species in the crystal's unit cell.

[I2 is accepted as standard by IUCr conventions, if by choosing I2 –over the much more common C2 centered monoclinic cell– the beta angle ends up being closer to 90 degrees, avoiding the cell to becoming too oblique. One can always reindex I2 into a C2 setting, and no one would question that SG C2 indeed possesses a 2-fold crystallographic axis].

Even though in solution at 1mM there is no indication of dimerization (MALS-SEC), in crystal (even more when solvent content is low), the effective concentration of protein can be much higher, and dimerization could be stabilizing 'active-like' conformation features in the protein.

Additional minor comments:

- From the Introduction (and Fig 1), I find the definition of HPT domains a little bit confusing. ChpT is not a proper example of HPT (pfam PF01627). His-containing phosphotransfer domains are monomeric, and display a different 4-helix bundle topology compared to the DHp-containing phosphotransferases (which are pseudo-HKs, like ChpT). There are numerous examples of HPTs, either alone (such as YPD1 from *S. cerevisiae*) or forming part of hybrid HKs (like within *E. coli* ArcB, or *B. pertussis* BvgS, etc etc), which indeed play key roles in many prokaryotic and eukaryotic phosphorelay pathways.

- Lines 55-56: there is more variation to this scheme, again think of the sporulation-control pathway in *B. subtilis*: the KinA-E HKs are not hybrid, and there's no HPT proteins involved (Spo0B is a DHp-containing pseudo-HK instead).

- Lines 357-370: an alternative explanation for the modeled His-Asp long distance in the CckA DHp/Rec interface, is that the REC domains might bind differently. This cannot be ruled out, and by mimicking the ChpT/CtrA crystal structure unanticipated biases may be extrapolated.

- Besides the comparison to other RECinter proteins that have been analyzed in terms of the effect of phosphorylation in modulating their conformations (see above), I would suggest the authors to consider also in the Discussion the following point: if there are no phosphorylation-dependent changes in phosphorelay intermediary RECs, and the fold is constitutively 'active', what derived consequences could be expected, e.g. in terms of binding constants to upstream and downstream His-containing DHp/HPT domains/proteins?

- Perhaps a more thorough checking for typos would be good (e.g. line 358, Willet's paper reference is missing; line 432, a parenthesis seems to be missing; line 781 in Fig 2's legend, the sentence doesn't seem to end properly; line 907 in Fig 9's caption, the full co-variation matrix is referred to in Fig. S8, whereas it should probably read Fig S6.; etc etc)

Reviewer #2 (Remarks to the Author):

This manuscript describes a newly solved structure of CckA(Rec) at high resolution reproducing a canonical fold of these Rec domains with some disordered part around helix 4. The disorder of helix 4 constitutes a deviation from other Rec domains such as CtrA or ChpT, but has been found in PhoB. Even less folds has been observed indicating stability of the REC fold. The structure is found in the active conformation without adding for example BeF3-. Extensive comparison with other Rec domains is made. Investigation of the conformation of CckA(Rec) with NMR confirms the fold seen in X-ray and the addition of BeF3- does not change the secondary structure and from heteronuclear NOEs also no change in mobility. Yet, Fig. S4 indicates quite big chemical shift changes upon binding. Did the authors try to rationalize those shift changes beyond just analyzing secondary shifts, for example potentially side chain reorganization which might also be essential for the active conformation?

The intermolecular phosphate transfer is modelled with the given structures of CckA domain and does not produce clashes. Did the authors consider doing titration experiments with these domains to experimentally back up the proposed interaction sites?

Further modelling is done for the phosphorelay based on the known structures and sequence comparison with CckA(Rec) and CtrA(Rec) and hypotheses about the functional specialization are produced. In addition a motif called "FATGUY" is analyzed in the various Rec domains. Further, co-variation between DHp and Rec is discussed.

The manuscript consists of a novel structure that reproduces the Rec fold with some variations seen in other Rec proteins. The rest are mainly comparisons of the specific Rec domain structure with other structures and this part reads more like a review than a research paper. Also, regarding interactions between the discussed domains this is mainly inferred from the individual structures but there is no experimental back up.

Thus I wonder (and leave to the editor) whether this manuscript should be condensed to the novel finding and the comparison with other structures rather to a review paper for two-component specialists.

Reviewer #3 (Remarks to the Author):

The manuscript by Brüderlin et al. reports a very high-resolution structure of the C-terminal receiver (REC) domain of the hybrid sensor histidine kinase CckA from the bacterium *Caulobacter crescentus*. This histidine kinase plays a key role in the regulation of the cell cycle in that bacterium and the analysis of the 1.25 Å-resolution structure of its REC domain allowed the authors to provide a

detailed structural description of the CckA-mediated His-Asp-His-Asp phosphorelay. In addition, a comparison of this REC domain with N-terminal REC domains of two-component response regulators allowed the authors to identify certain distinguishing features of the two types of REC domains. This is a very thorough work that provides a long-sought structural view at the mechanisms of two-component signaling phosphorelays. It substantially advances our understanding of two-component signal transduction and opens new avenues for in-depth exploration of bacterial regulatory mechanisms. The whole paper is well-planned, well-written, and well-illustrated. The methods are described in sufficient detail. The high-resolution structure solved in this work is already available in the PDB (PDB: 6tne).

I have only some minor comments for the authors.

Minor comments

L. 25. Change "receiver" to "receiver (Rec)". For clarity, I would suggest rendering the domain name in caps, i.e. REC, throughout the manuscript.

L. 30 and elsewhere. I am afraid that abbreviating the REC domain of CckA as CckA^{Rec} might be confusing for some. It might make sense to display various REC domains as REC with a subscript indicating the source, e.g. changing CckA^{Rec} to REC_{CckA}, PhoB^{Rec} to REC_{PhoB} and so on.

L. 34. 'Hall mark' should be 'hallmark'

L. 36. Change "CckA DHp/Rec domains" to "DHp and REC domains of CckA"

L. 96-106. Add the references for 2iyn, 6qrl, 2ayx and 3q15.

L. 163. Remove ",Phosphate".

L. 227-228 (also, 245-246). "For a comparative analysis, Hidden Markov Models (HMM) based on the sequences of CckAREC and CtrAREC were created." Did you mean sequences homologous to CckAREC and CtrAREC?

L. 239. Please spell out the "4-5-5 face".

L. 358. Change "{Willet}" to Ref. 17.

L. 575-578, Figure 7. How many sequences have been aligned for each of the sequence logos?

L. 599. Change "to thanks to" to "to thank"

L. 768. Change "classical" to DNA-binding".

Legend to Fig. 9. Please provide the numbers of HHK sequences used for the co-variation analysis in panels A and B.

Reviewer #1 (Remarks to the Author):

Brüderlin and collaborators report Rx crystallographic and NMR data studying the intermediary REC domain of the hybrid his-kinase (HHK) CckA from *Caulobacter crescentus*. They also perform clever sequence bioinformatics analyses, extending their observations to understand more general principles that might discriminate between intermediary Asp-containing response regulators (in phosphorelay pathways) and terminal RRs (the ones producing TCS and phosphorelay pathways' output effects).

The manuscript is well written and, supported on a beautiful set of illustrations, the messages are indeed very clear.

The knowledge gap the authors have set out to fulfill is based on the fact that intermediary REC domains have received less attention compared to terminal ones. While the goal is interesting, there is not a clear hypothesis behind the study.

Since it was known that Rec_{inter} domains are not using the 4-5-5 face for interacting with their phosphorelay partners (see the Spo0F/B and ChpT/CtrA complexes), we hypothesized that the allosteric mechanism evolved in Rec_{term} domains to change the structure of the 4-5-5 face would not be needed. This hypothesis is corroborated by our presented findings showing that the CckA^{Rec} structure indeed does not show any allosteric response to pseudo-phosphorylation.

In particular, the authors don't seem to exploit previous work in the field, which I believe is relevant, and that would've helped in defining a clearer set of premises and hypotheses for their own work. They do refer to results reported by the Laub group (Capra et al doi:10.1111/mmi.12064), nicely showing now, that HHKs do reveal amino acid coevolution among their tethered kinase and receiver domains (interestingly contrasting to Capra and coworkers' hypothesis, likely due to the use of smaller sequence datasets then available).

Looking again into this contrasting finding, we now offer an additional possible reason:

line 522ff: Lower specificity of the intramolecular recognition in HHKs compared to cognate HK/RR pairs has been reported and attributed to the relaxed constraints for phosphotransfer due to the high local concentration of the covalently attached Rec domain^{33,34}. **In both studies, covariation between HHK DHp and Rec residues was found to be much lower compared to canonical (non-tethered) pairs.** Here, by being able to analyze a vast number of sequences, we have found very significant

covariation between the tethered DHP and Rec domains of HHKs (Fig. 9). It would be interesting to see, **whether covariation between cognate DHP/Rec from separate proteins turns out to be even more pronounced, when using a similarly large database. Also, we note that most HHK sequences available at the time of the earlier studies^{33,34} covered a wide phylogenetic range, which raises the question whether all those proteins use equivalent contact points for the intra-molecular recognition, a pre-requisite for a covariation analysis, whereas in our study we used a more closely related group of sequences. In any case, it should be considered that an HHK Rec domain not only has to recognize its covalently attached DHP domain, but also its specific downstream HPT partner, with latter interaction subject to the same evolutionary constraints as in canonical HK/RR systems.**

However, this is not discussed in the context of the reduced constraints in the specificity of these domains' binding that early reports had shown (such as in the 2013 paper by Townsend et al. doi:10.1073/pnas.1212102110, which is not cited).

We thank the referee for directing us to the Townsend et al. (2013) paper, which indeed provides very similar results as Capra et al. (2012) and was published independently about the same time as the Capra paper. We have replaced ref. 32 (review by Capra et al., Annual Review, 2012) by Townsend et al., 2013 (ref. 34).

Most importantly, the 2021 report by Ortet et al (doi:10.1038/s41598-021-91260-w) has also been overlooked, even though that paper, using similar kinds of co-variance analyses in HHKs, had already shown quite convergent observations and conclusions to the ones the authors are now presenting. For instance, what Brüderlin et al conclude here in lines 384-393, goes very much in the same direction as the Ortet et al paper did, in the sense that inter-domain specificity has been selected for during evolution, also in intra-molecular HHK associations.

In this paper, cross-talk between kinase and Rec domains belonging to phylogenetically related HHKs of the same organism (GacS–GacA multikinase network of *Pseudomonas brassicacearum*) is investigated both bioinformatically (interaction energies based on coevolution) and experimentally. Though a large number of HKs and Rec domain sequences (from the P2CS.org database) were used to construct the evolutionary model as a basis for the interaction energies, HHKs were not selected or treated specially for this as far as we understand and we don't find

pertinent new findings about their evolutionary constraints. Thus, we found that this paper is not of relevance to our study.

Besides the RECinter-DHp/HPt specificity analysis, the effect of phosphorylation in stabilizing/triggering conformational changes of RECinter domains had also been analyzed in a number of papers. I will mention one by Gardino et al in 2003 (not referenced here), where extensive analysis of the *B. subtilis* Spo0F (a monodomain RECinter protein, within the phosphorelay pathway that regulates sporulation) was performed by NMR.

In fact, we found Spo0F to be the closest structural homolog of CckA^{Rec}, see line 125ff. We are aware that there are a few structures of Rec_{inter} domains in the PDB. These are listed in Supplementary Fig. 5b and include Spo0F. We classified all of them with respect to 5 distinct HMM profiles. We have changed the title of Supplementary Fig. 5 to emphasize that only Rec_{inter} domains are listed.

Sequence logos for five distinct **Rec_{inter}** groups and correlation (E-values) of selected sequences against derived HMM motifs.

Gardino et al do identify conformational variations associated to phosphorylation, and also focus on helix alpha4 –which they see by NMR– already pinpointing that it is a less stable helix.

We are thankful for the suggestion to discuss Spo0F in more detail, since there are both native and pseudo-phosphorylated structures available. However, it turns out that the conformational changes proposed by Gardino et al. (2003) were not confirmed later by crystallography. Thus, we added this paragraph to the discussion and added a new Supplementary Fig. 7.

line 639ff: Spo0F from *B. subtilis* is another well-studied single-domain Rec_{inter} protein. Although in an initial NMR study³⁹ significant shifts of helices α_1 , α_3 , α_4 were reported upon BeF₃⁻ modification, this was not confirmed later by the crystallographic study of Varughese and coworkers³¹. Surprisingly at that time, the BeF₃⁻-bound Spo0F structure superimposed closely with native Spo0F (both determined in complex with Spo0B)⁴⁰ (Supplementary Fig. 7).

I also wonder whether Brüderlin et al also observe this helix α_4 by NMR now (it appears to be the case, according to lines 143-144). Maybe the helix is there in the crystal structure but given its higher flexibility, its electron density is weaker? (see

below, but maybe a more complete/accurate model-building job, resulting in lower R factors, could eventually reveal helix $\alpha 4$'s density?).

To which degree $\alpha 4$ is defined is functionally not crucial, as far as this segment does not change conformation. We found low helical propensity for this helix by NMR and no interpretable electron density in the crystallographic map (also after re-refinement, see below). This may point to a less significant element unless there would be increased order upon BeF₃- titration which we don't see in our NMR experiment.

Overall then, the new insights coming from this manuscript are not that clear to me.

We are glad that our “**messages** are indeed very clear” as the referee states at the beginning, but the main message or **insight** seems not clear enough. Actually, the main value of this study, in our opinion, would be the corroboration of the hypothesis outlined at the beginning. In the abstract, the main insight is summarized as “BeF₃ binding does not alter secondary structure nor the oligomeric state, indicating the absence of allosteric changes, the hallmark of RRs.” Furthermore, we hope that the functional classification of Rec domains Rec_{term}, Rec_{inter}) will open a new venue for further structural and bioinformatic studies.

Major comments:

- Refinement at this resolution (which is well within atomic resolution; particularly considering that this crystal most likely diffracted to higher resolution than what the authors decided to collect, according to highest resolution shell statistics), requires special attention to standard protocols :
 - o Rwork and Rfree seem quite high (the validation report picks this up immediately of course);
 - o B-factors should most probably be refined anisotropically (if a significant drop in Rfree is achieved by doing so);
 - o several clashes that are now observed in the validation report, involve water molecules: they could be indications of double/multiple conformations for several side chains (typically observed at atomic resolution);
 - o as it is now, the model includes quite low a number of ordered water molecules. Especially considering that the average B factor is quite low. With the figures reported on Table 1, I would expect that for a ~120-residues protein, we would be able to see

in the range of >200 water molecules. For this purpose, it would be important to check how the difference Fourier map looks like at this stage of refinement, in terms of unaccounted peaks (i.e. how many peaks does the mFo-DFc map show, e.g. countoured at 3.5σ , and what heights do those peaks display?). Such analysis would objectively gauge whether there are significant features that could still be (re)modelled. None of these protocols, standard for atomic resolution data, are mentioned in the Methods section, so I assume they were not considered. These comments are not mere technicalities, but rather anchored on the important concept that phases improve as the model gets more complete and accurate. Better phases impact on the quality of the whole map, including any regions that might so far be less well defined (the $\beta 4$ - $\beta 5$ linker? and all others).

We would like to thank the reviewer for the suggestion to further improve the structure by making full use of the very high resolution. Indeed, anisotropic B-factor refinement and optimization of the X-ray/stereochemistry and ADP weights significantly dropped the R_{free} drastically from **22.7% to 18.8%** and R_{work} from **20.6% to 16.5%**. We have also removed clashing waters and modeled R639 as two alternative conformations (apart from F670). With the improved phases, we have reattempted to build the $\beta 4$ - $\beta 5$ linker. There is density in this region, but we were unable to model the residues confidently into the blobs, therefore we decided to leave them empty.

Obviously, we also checked the difference Fourier map (mFo-DFc map countoured at 3.5σ). The identified 50 peaks (10.69 - 3.63σ) all belong to the corresponding blobs for $\beta 4$ - $\beta 5$. 10 more peaks corresponding to water molecules were identified. The relatively low number of water molecules (103) is probably due to the small solvent volume (30 %, $V_{\text{M}} = 2.0 \text{ \AA}^3/\text{Da}$). Please note that we have deposited the updated coordinates in the PDB and the statistics Supplementary Table 1.

I2 is an infrequently used (body-centered) setting of monoclinic space groups: by definition it possesses a 2-fold (dyad) axis. Please correct (lines 108-111; this misconception is also implicit in line 350, I suspect). Hence, it does seem relevant to analyze whether this dyad generates a dimeric species in the crystal's unit cell.

Indeed, we inadvertently made a mistake here, which we have now corrected. But there are no crystallographic dimers with significant interface area present in the crystal. The statement of line 474 is still fine. The other pertinent sentence (line 128ff) now reads

The asymmetric unit of the CckA^{Rec} crystals contains one Rec monomer and no significant interfaces are formed within the crystal lattice (all interface areas < 275 Å²).

I2 is accepted as standard by IUCr conventions, if by choosing I2 –over the much more common C2 centered monoclinic cell– the beta angle ends up being closer to 90 degrees, avoiding the cell to becoming too oblique. One can always reindex I2 into a C2 setting, and no one would question that SG C2 indeed possesses a 2-fold crystallographic axis.

We were aware of the ambiguity in space-group assignment. In fact, we followed POINTLESS (CCP4) suggestion for I2, which yields a β -value of 94.36° as opposed to C2 with a β -angle of 166.5.

Even though in solution at 1mM there is no indication of dimerization (MALS-SEC), in crystal (even more when solvent content is low), the effective concentration of protein can be much higher, and dimerization could be stabilizing ‘active-like’ conformation features in the protein.

The argument is well taken. However, no tight dimers or other assemblies, which may be responsible for inducing a particular Rec conformation, are present in the crystal (please see above).

Additional minor comments:

From the Introduction (and Fig 1), I find the definition of HPt domains a little bit confusing. ChpT is not a proper example of HPt (pfam PF01627). His-containing phosphotransfer domains are monomeric, and display a different 4-helix bundle topology compared to the DHp-containing phosphotransferases (which are pseudo-HKs, like ChpT). There are numerous examples of HPts, either alone (such as YPD1 from *S. cerevisiae*) or forming part of hybrid HKs (like within *E. coli* ArcB, or *B. pertussis* BvgS, etc etc), which indeed play key roles in many prokaryotic and eukaryotic phosphorelay pathways.

We agree that ChpT does not carry an HPt (pfam PF01627) domain, instead it is a pseudo-HK composed of a DHp domain and a degenerated AMP-binding (CA) domain

with the DHp domain full-filling the same phosphotransferase function as HPT domains in other phosphorelays. In Fig. 1b, we have chosen a schematic pseudo-HK (DHp-CA) as a representative for the histidine phosphotransferase, obviously to connect straightforwardly with the detailed panel c.

To be more general, we now refer to HPT in the legend to Fig. 1b:

(b) Multistep phosphorelay consisting of a hybrid histidine kinase (HHK), a histidine phosphotransferase and a response regulator. **The histidine phosphotransferase can either be an HPT protein or a pseudo-HK, as shown.**

Furthermore, we have changed the relevant sentence (lines 58ff) to

Multi-step phosphorelays are more complex involving at least one additional Rec domain and a **histidine phosphotransferase (either an HPT protein or a pseudo-HK)**, both of which act as intermediaries in the phosphotransfer^{1,4}.

Also, we changed this sentence (line 63ff)

(1) Rec domains that act as intermediaries in the phosphotransfer chain by shuttling the phosphoryl between **the active histidines of an HHK and a histidine phosphotransferase** (hereafter referred to as Rec_{inter})

Lines 55-56: there is more variation to this scheme, again think of the sporulation-control pathway in B subtilis: the KinA-E HKs are not hybrid, and there's no HPT proteins involved (Spo0B is a DHp-containing pseudo-HK instead).

We have changed the sentence, see also above.

Multi-step phosphorelays are more complex involving at least one additional Rec domain and a **histidine phosphotransferase (either an HPT protein or a pseudo-HK)**, both of which act as intermediaries in the phosphotransfer^{1,4}.

Lines 357-370: an alternative explanation for the modeled His-Asp long distance in the CckA DHp/Rec interface, is that the REC domains might bind differently. This cannot be ruled out, and by mimicking the ChpT/CtrA crystal structure unanticipated biases may be extrapolated.

We agree with the reviewer that we can not rule out that the CckA Rec domain might bind slightly different to its DHp domain as in the ChpT/CtrA(Rec) template complex. We have

added a new figure supplementary Fig. 7 that demonstrates such variation between the ChpT/CtrA(Rec) and the Spo0F/Spo0B complexes. We have added this caveat to the text:

502ff: In CckA, such interactions could involve the Y of the FATGUY motif and subsequent residues from the Rec β 4 - β 5 linker and may contribute to affinity, but also to α 1' bending for efficient phosphotransfer. **Alternatively, the relative orientation of the DHp and Rec domains in CckA may not be exactly the same as in the ChpT/CtrA(Rec) template complex. This notion is supported by the observation that the relative domain orientation within the experimental Spo0F/B and ChpT/CtrA complexes differs by about 30 deg although the same interface is used (Fig. S7).**

Besides the comparison to other RECinter proteins that have been analyzed in terms of the effect of phosphorylation in modulating their conformations (see above), I would suggest the authors to consider also in the Discussion the following point: if there are no phosphorylation-dependent changes in phosphorelay intermediary RECs, and the fold is constitutively 'active', what derived consequences could be expected, e.g. in terms of binding constants to upstream and downstream His-containing DHp/HPt domains/proteins?

This touches on the directionality of phosphotransfer in phosphorelays. One could imagine that allosteric changes induced by phosphorylation would increase the affinity of the RECinter domain to the downstream DHp/HPt protein and in this way favour forward over back transfer. To our knowledge no such effect has been described for phosphorelays. For canonical TCSs, it has been reported that there is minimal Rec(term)-P to HK back transfer, which has been linked to a dissociative as opposed associative phosphorylation reaction (Trajtenberg et al, Elife 2016).

Many phosphorelays, including the CckA-ChpT-CtrA, have been shown to be bifunctional, i.e. phosphorylating as well as dephosphorylating its Rec(term) domain (depending on differential regulation of the kinase, see Dubey et al., Sci. Adv. 2016). We speculate that at equilibrium the entire phosphorelay will get populated or depopulated, such that the individual Kd's between the members will be of lesser importance and that their modulation by Rec(inter) phosphorylation due to allosteric changes would functionally not be required (i.e. it would work well with a constitutively active Rec(int) structure). In our paper, we want to refrain from such farther reaching discussions, considering that they would be very speculative given the scarcity of pertinent data for multi-step phosphorelays.

Perhaps a more thorough checking for typos would be good (e.g. line 358, Willet's paper reference is missing; line 432, a parenthesis seems to be missing; line 781 in Fig 2's legend, the sentence doesn't seem to end properly; line 907 in Fig 9's caption, the full co-variation matrix is referred to in Fig. S8, whereas it should probably read Fig S6.; etc etc)

We have corrected all typos and mistakes in the text using tracked changes.

Reviewer #2 (Remarks to the Author):

This manuscript describes a newly solved structure of CckA(Rec) at high resolution reproducing a canonical fold of these Rec domains with some disordered part around helix 4. The disorder of helix 4 constitutes a deviation from other Rec domains such as CtrA or ChpT, but has been found in PhoB. Even less folds has been observed indicating stability of the REC fold. The structure is found in the active conformation without adding for example BeF₃⁻. Extensive comparison with other Rec domains is made. Investigation of the conformation of CckA(Rec) with NMR confirms the fold seen in X-ray and the addition of BeF₃⁻ does not change the secondary structure and from heteronuclear NOEs also no change in mobility.

Yet, Fig. S4 indicates quite big chemical shift changes upon binding. Did the authors try to rationalize those shift changes beyond just analyzing secondary shifts, for example potentially side chain reorganization which might also be essential for the active conformation?

The chemical shift perturbations (CSPs) caused by BeF₃⁻/Mg⁺⁺ binding (derived from the spectra of Supplementary Fig. 4) are shown in Fig. 4c. Almost all of them are below 0.4 ppm consistent with the absence of significant conformational changes. These changes are much smaller than e.g. observed for NtrC^{Rec}, a Rec_{term} domain that undergoes reorganization of the 4-5-5 face upon activation (Review-only-Figure 1).

Review-only-Figure 1. CSP analysis of published data on NtrC (Pontiggia et al., Nat Commun 6, 7284 (2015)) as a comparison for our current data. CSPs were computed between the chemical shift values of the $\text{BeF}_3^-/\text{Mg}^{++}$ treated and the apo sample, using the same formula as for the CckA Fig.4C. (see Methods). Data were taken from the BMRB.io database, entries 30024 and 30025.

The improved Fig. 4D shows the CSPs of CckA^{Rec} mapped onto the crystal structure. The few CSPs > 0.4 ppm cluster around the BeF_3^- binding site. These perturbations can be fully rationalized by the addition of negatively charged molecules to these sites, which will change the electronic surroundings of the nearby nuclei substantially, and thus may also include some limited local structural changes. It appears thus not necessary to study side-chain orientations. We have added the pertinent text in the manuscript:

line 203ff: All residues with strong or medium perturbation cluster around the phosphorylation site (Fig. 4D), with residues 648 – 658, located in $\beta 4$ and the beginning of the $\beta 4 - \beta 5$ linker, representing the largest continuous stretch.

The intermolecular phosphate transfer is modelled with the given structures of CckA domain and does not produce clashes. Did the authors consider doing titration experiments with these domains to experimentally back up the proposed interaction sites?

We think that an experimental verification of the proposed interaction sites would be a major project and out of scope. One could perform NMR experiments utilizing the labeled and assigned Rec domain to probe for chemical shift perturbations (CSPs)

upon complex formation. But this would identify only coarsely Rec residues that are directly or indirectly interacting, but not interaction pairs. Obviously, we tried to crystallize the complex, but without success. On the other side, the covariation analysis as reported here gave rather clear results consistent with the modelled complex. The major power of covariation analysis is that it not only identifies the residues that involved in the interaction, but also their partners (i.e. interaction pairs), see Fig. 9.

Further modelling is done for the phosphorelay based on the known structures and sequence comparison with CckA(Rec) and CtrA(Rec) and hypotheses about the functional specialization are produced. In addition a motif called "FATGUY" is analyzed in the various Rec domains. Further, co-variation between DHp and Rec is discussed.

The manuscript consists of a novel structure that reproduces the Rec fold with some variations seen in other Rec proteins. The rest are mainly comparisons of the specific Rec domain structure with other structures and this part reads more like a review than a research paper.

Only the first chapter of the Results reports on the high-resolution structure of CckA^{Rec} and comparisons with other Rec structures, but there are six more chapters with original research (NMR, modelling, bioinfo) and a novel functional classification of Rec domains following. In our view, this is clearly a research paper and not a review.

Also, regarding interactions between the discussed domains this is mainly inferred from the individual structures but there is no experimental back up.

Please see our response above.

Thus I wonder (and leave to the editor) whether this manuscript should be condensed to the novel finding and the comparison with other structures rather to a review paper for two-component specialists.

We hope that the editor will agree with us that this paper represents genuine, novel research and that the discussion is tightly connected to the novel findings and not to general aspects of two-component systems.

Reviewer #3 (Remarks to the Author):

The manuscript by Brüderlin et al. reports a very high-resolution structure of the C-terminal receiver (REC) domain of the hybrid sensor histidine kinase CckA from the bacterium *Caulobacter crescentus*. This histidine kinase plays a key role in the regulation of the cell cycle in that bacterium and the analysis of the 1.25 Å-resolution structure of its REC domain allowed the authors to provide a detailed structural description of the CckA-mediated His-Asp-His-Asp phosphorelay. In addition, a comparison of this REC domain with N-terminal REC domains of two-component response regulators allowed the authors to identify certain distinguishing features of the two types of REC domains. This is a very thorough work that provides a long-sought structural view at the mechanisms of two-component signaling phosphorelays. It substantially advances our understanding of two-component signal transduction and opens new avenues for in-depth exploration of bacterial regulatory mechanisms. The whole paper is well-planned, well-written, and well-illustrated. The methods are described in sufficient detail. The high-resolution structure solved in this work is already available in the PDB (PDB: 6tne).

I have only some minor comments for the authors.

We appreciate the reviewer for his/her insightful comments and the recognition of our work. We thank the referee for pointing out some typos and mistakes, which we have corrected now. Below is a point-by-point response to the reviewers' comments and concerns.

Minor comments

L. 25. Change "receiver" to "receiver (Rec)".

We have changed it.

For clarity, I would suggest rendering the domain name in caps, i.e. REC, throughout the manuscript.

We would like to stick to “Rec”, since this abbreviation is well introduced and we have used it throughout our work. (REC in capital letters looks like an acronym as e.g. HTH domain).

L. 30 and elsewhere. I am afraid that abbreviating the REC domain of CckA as CckA^{Rec} might be confusing for some. It might make sense to display various REC domains as REC with a subscript indicating the source, e.g. changing CckA^{Rec} to REC_{CckA}, PhoB^{Rec} to REC_{PhoB} and so on.

We think the abbreviation for CckA^{Rec} is not confusing. We followed the same style of abbreviation for Rec domains from different sources in the text and the figures and maintained consistency in the paper. Changing abbreviation at this point would be substantial work (need to change also in figures) and we think this might be a matter of taste.

L. 34. ‘Hall mark’ should be ‘hallmark’

We have corrected it.

L. 36. Change “CckA DHp/Rec domains” to “DHp and REC domains of CckA”

We have corrected it.

L. 96-106. Add the references for 2iyn, 6qrl, 2ayx and 3q15.

Thanks for noticing that, we added the references to the corresponding PDBs code.

L. 163. Remove ",Phosphate".

Done.

L. 227-228 (also, 245-246). “For a comparative analysis, Hidden Markov Models (HMM) based on the sequences of CckA^{Rec} and CtrA^{Rec} were created.” Did you mean sequences homologous to CckA^{Rec} and CtrA^{Rec}?

Yes.

L. 239. Please spell out the “4-5-5 face”.

4-5-5 is spelled out as $\alpha 4\text{-}\beta 5\text{-}\alpha 5$ and added in the manuscript.

L. 358. Change “{Willet}” to Ref. 17.

Done.

L. 575-578, Figure 7. How many sequences have been aligned for each of the sequence logos?

We have added the information to the legend of the figure.

The CckA^{Rec} and CtrA^{Rec} logos are based on the alignment of 132 and 571 sequences, respectively.

L. 599. Change “to thanks to” to “to thank”

We have removed the whole sentence as Firas Fadel became an author.

L. 768. Change “classical” to DNA-binding”.

We have changed it.

Legend to Fig. 9. Please provide the numbers of HHK sequences used for the co-variation analysis in panels A and B.

We have used 8067 sequences for our co-variation analysis shown in panel a, as mentioned in the legend of Fig S6 (Line 1389ff). The results shown in panel b are derived from the co-variation analysis of panel a.

REVIEWER COMMENTS

Reviewer #1 (Remarks to the Author):

I wish to thank the authors for their thorough responses and for the additional work, which have overall improved the ms.

Quite clarifying to me, I strongly suggest for some of these replies to be included in the text of the manuscript:

1- The authors now formulate their central hypothesis, which is mechanistic. Namely that the allosteric mechanism evolved in RecTerminator domains to change the structure of the 4-5-5 face is not needed.

Relevant sections of the article should state this hypothesis explicitly, so that readers are aware early on, especially in the Abstract.

Also in the Introduction and the pertinent Results subsection of course.

In this way the article changes substantially: from being only descriptive (as it now reads, it seems that because little is known about RecTerminator domains, the authors decided to study the C-terminal Rec domain of CckA by Rx and NMR), it becomes one that dives into the distinctive mechanism(s) of HHKs signal-transmission.

2- Given that the significance of the FATGUY motif is still not resolved, I would suggest avoiding mentioning that in the Abstract, and also shortening its analysis in the Discussion (in favor of the main messages, see further below).

3- About the authors' reply "We are glad that our "messages are indeed very clear" as the referee states at the beginning, but the main message or insight seems not clear enough" (bold fonts included): should I take it as an ironic response?

I am sorry if the authors got this wrong, but those comments were both candid and honest. I still stand by them, and they are not mutually exclusive.

Indeed, the phrase for that main discovery was included in the first version's Abstract, but to my understanding, then gets lost within the paper.

A proof of that is that the sub-section "Preformed CckARec active site", is the shortest among all the sub-sections in the Discussion. However, this is the one referring to THE central discovery of the paper (see point below about Discussion).

4- The Introduction of the article should end by highlighting the main results(s) of the article, and the novelty of the insights. Please rephrase appropriately to include these two.

5- With hypothesis and main discoveries now better highlighted, the question is whether this is indeed an extended property among Recinter domains in HHKs. A one-case scenario wouldn't be so interesting.

The authors are indeed cautious in referring all their analyses to this particular Caulobacter protein. I recommend paying more attention in the Discussion, to put these discoveries in the broader context: building on comparative analyses of CckARec with other HHK RECs (sequence and structure-wise), what can be learned/predicted concerning the molecular bases of this lack of changes in the structure of the 4-5-5 face? Is this a general phenomenon or a CckA-specific curiosity?

Probably reducing the length of other sub-sections in the Discussion could be worth, so that main messages remain central. Actually, in lines 463-464, what do you mean by the lysine (K673) having been proposed to switch from an outward to an inward orientation in signaling? No reference is indicated, and this is not obvious to me. Please double-check and correct.

6- The conclusion that phosphorylation does not produce any structural modification in the REC domain, is really important, and goes against the typical scenario for autonomous REC domains in RRs for many TCSs studied so far (independently of them being REC-only or not).

Two points that I would like the authors to double-check to further support their claim:

* The authors say there is a water molecule sitting right in place of a typical Mg²⁺ cation. The atomic number of the metal and of a H₂O molecule (one can't distinguish separate hydrogens at this resolution) are identical, so the Mg/water distinction cannot be made on the basis of electron density peak height. Please check and report on the binding geometry and the bond lengths of this alleged water to the atoms to which it is bound. We can't realize this from fig 3b right now (not sure if the authors are showing all the bonds there)

Is an octahedral architecture found? (with 4 equatorial and 2 axial atoms bound); and, most importantly, are bond lengths tending to refine to ~2.1Å or to ~2.7Å. These data will be truly telling, to distinguish a cation from a water, and should be mentioned.

The rationale behind this doubt, is that the protein does contains Mg²⁺ in the last purification buffer, and this is true both for the Rx studies as well as for the NMR analyses. We wouldn't want Mg to be bound and explain an "active state"-like configuration.

* Even if “in crystal” dimerization (via the crystallographic axis) can now be ruled out, are there any other crystal contacts that could otherwise preclude the protein to adopt a more relaxed/open configuration (typical of the inactive state)?

These would be crystal contacts either near the phosphorylatable Asp reaction center, or at the 4-5-5 face.

Phe670 is indeed adopting both active and inactive conformations; but further relaxation should for instance allow the $\alpha 1$ - $\beta 1$ loop to move away from the reaction center (e.g. Glu577 and Asp578 would normally move away from the site in the absence of phosphoryl-Mg²⁺ bound) etc Are there crystal contacts that would preclude these movements?

7- It is very good to see that refinement/modeling improved substantially. If many of those strong Fourier difference peaks appear at the “missing” $\beta 4$ - $\beta 5$ linker, I believe this deserves being described, for the readers to be aware.

Lines 91-92 should convey a better description, using some of the words the authors use now to respond to the reviewers (in the line of “There is density in this region, but not clear enough to model the linker segment reliably (Supp Fig X) ” etc). A supplementary panel will be helpful, showing the density, and hence enriching the image readers will get, beyond the simple absence of any model, which is what Fig 2a shows.

8- Good to know that the I2 setting was well chosen and that the low number of ordered waters is probably due to the low solvent content.

Relevant literature is also better credited now, allowing to fit this nice piece of work into a broader context.

Reviewer #2 (Remarks to the Author):

The authors have addressed all points that I raised.

Reviewer #1 (Remarks to the Author):

I wish to thank the authors for their thorough responses and for the additional work, which have overall improved the ms.

Quite clarifying to me, I strongly suggest for some of these replies to be included in the text of the manuscript:

1- The authors now formulate their central hypothesis, which is mechanistic. Namely that the allosteric mechanism evolved in Rec_{term} domains to change the structure of the 4-5-5 face is not needed.

Relevant sections of the article should state this hypothesis explicitly, so that readers are aware early on, especially in the Abstract.

Also in the Introduction and the pertinent Results subsection of course.

In this way the article changes substantially: from being only descriptive (as it now reads, it seems that because little is known about Rec_{inter} domains, the authors decided to study the C-terminal Rec domain of CckA by Rx and NMR), it becomes one that dives into the distinctive mechanism(s) of HHKs signal-transmission.

We now state this hypothesis in the Abstract:

In contrast, multi-step phosphorelays comprise at least one additional Rec (Rec_{inter}) domain that acts as an intermediary for phosphoryl-shuttling probably not requiring allosteric changes at the remote 4-5-5 face.

and in the Introduction:

Previously, it has been shown that Rec_{inter} domains are not using the 4-5-5 face for interacting with their phosphorelay partners^{17,31}, therefore we hypothesized that the allosteric mechanism evolved in Rec_{term} domains to change the structure of the 4-5-5 face would not be needed.

In the Results we merely want to present the facts, but in the Discussion this main finding is stressed again (see below).

2- Given that the significance of the FATGUY motif is still not resolved, I would suggest avoiding mentioning that in the Abstract, and also shortening its analysis in the Discussion (in favor of the main messages, see further below).

We'd like to keep the FATGUY discussion in the paper. The finding that this motif is found not only in single-Rec proteins, but also in HHKs, is relevant to our general Rec_{inter} analysis. Furthermore, we think it's important to point out that published structural models (derived from NMR data) and the resulting activation mechanisms are flawed by not taking into account that the K-P peptide bond (following the active K) is always in *cis*. We hope that this will prompt further investigations in the community.

3- About the authors' reply "We are glad that our "messages are indeed very clear" as the referee states at the beginning, but the main message or insight seems not clear enough" (bold fonts included): should I take it as an ironic response?

I am sorry if the authors got this wrong, but those comments were both candid and honest. I still stand by them, and they are not mutually exclusive.

Indeed, the phrase for that main discovery was included in the first version's Abstract, but to my understanding, then gets lost within the paper.

A proof of that is that the sub-section "Preformed CckARec active site", is the shortest among all the sub-sections in the Discussion. However, this is the one referring to THE central discovery of the paper (see point below about Discussion).

We now understand better the referee's concern. It appears that he/she finds the hypothesis that Rec_{inter} domains don't experience allosteric changes upon phosphorylation a rather interesting one and that this hypothesis is supported by our findings. The referee would like to see the paper more tailored to this "main discovery". In response to comment 1 we have altered the Abstract and the Introduction.

Regarding the Discussion, we have expanded the "Preformed CckARec active site" sub-chapter (now titled "Lack of allosteric response to modification"). It now ends with

In summary, unlike the well-known phosphorylation-induced allosteric response at the 4-5-5 face in Rec_{term} domains¹⁸, there is no evidence that changes of similar magnitude take place in CckA^{Rec} and the same can be expected for its orthologs. Since mainly $\alpha 1$ is involved in the interaction with the phosphorelay partners (see e.g. Figs. 5, 9), changes at the distant 4-5-5 face would probably have no effect on phosphotransfer and would therefore be dispensable.

and have inserted another sub-title "Conclusion" to the last paragraph, coming back to the passive role of Rec_{inter} domains.

4- The Introduction of the article should end by highlighting the main results(s) of the article, and the novelty of the insights. Please rephrase appropriately to include these two.

Since our paper comprises several findings, we report various “main” results at the end of either of the last two paragraphs. We don’t find it necessary to stress that these results are obviously novel.

5- With hypothesis and main discoveries now better highlighted, the question is whether this is indeed an extended property among Rec_{inter} domains in HHKs. A one-case scenario wouldn’t be so interesting.

The authors are indeed cautious in referring all their analyses to this particular Caulobacter protein. I recommend paying more attention in the Discussion, to put these discoveries in the broader context: building on comparative analyses of CckA_{Rec} with other HHK RECs (sequence and structure-wise), what can be learned/predicted concerning the molecular bases of this lack of changes in the structure of the 4-5-5 face? Is this a general phenomenon or a CckA-specific curiosity?

The HMM analyses of CckA^{Rec} and four other Rec_{inter} proteins shown in Fig. S5 were performed exactly with the motivation to extract features that are characteristic for general Rec_{inter} domains in general. Somewhat dissatisfying we did not find such a universal motif, which unequivocally would allow classification of a Rec domain to be of the intermediary type and we attribute this to Rec_{inter} domains not being a monophyletic group (see end of Discussion subchapter “Rec classification”). We are also very clear that our finding of a passive Rec domain refers to CckA_{Rec} orthologs only and that other Rec_{inter} domains (e.g. of ShkA) have to be tested for this mechanistic property.

Probably reducing the length of other sub-sections in the Discussion could be worth, so that main messages remain central.

We hope that with the latest changes the main messages are now more visible. We agree that some sub-chapters of the Discussion are rather long, but find that this is required to put our findings in the right context.

Actually, in lines 463-464, what do you mean by the lysine (K673) having been proposed to switch from an outward to an inward orientation in signaling? No reference is indicated, and this is not obvious to me. Please double-check and correct.

Here we refer actually to K102 of SdrG and the NMR study by Campagne et al., 2016. As discussed, the models have been determined under the constraints of a *trans* conformation for the 102 - 103 peptide bond (PDBs 5ieb, 5iej), while most likely it is *cis* as in all known Rec crystal structures. We have changed the sentence to

Thus, the proposed role of the lysine to switch from an outward to an inward orientation upon activation in SdrG has to be questioned¹⁰.

6- The conclusion that phosphorylation does not produce any structural modification in the REC domain, is really important, and goes against the typical scenario for autonomous REC domains in RRs for many TCSs studied so far (independently of them being REC-only or not).

Two points that I would like the authors to double-check to further support their claim: The authors say there is a water molecule sitting right in place of a typical Mg²⁺ cation. The atomic number of the metal and of a H₂O molecule (one can't distinguish separate hydrogens at this resolution) are identical, so the Mg/water distinction cannot be made on the basis of electron density peak height. Please check and report on the binding geometry and the bond lengths of this alleged water to the atoms to which it is bound. We can't realize this from fig 3b right now (not sure if the authors are showing all the bonds there)

Is an octahedral architecture found? (with 4 equatorial and 2 axial atoms bound); and, most importantly, are bond lengths tending to refine to ~2.1Å or to ~2.7Å. These data will be truly telling, to distinguish a cation from a water, and should be mentioned.

The rationale behind this doubt, is that the protein does contains Mg²⁺ in the last purification buffer, and this is true both for the Rx studies as well as for the NMR analyses. We wouldn't want Mg to be bound and explain an "active state"-like configuration.

We have redrawn all non-covalent bonds in Fig. 3b, now within a distance of 3,2 Å. All water molecules show non-covalent distances of > 2.8 Å, precluding the presence of a divalent cation. Upon further inspection we now realize that the cation binding site is in fact empty, with the nearest water at a distance of 1.7 Å to this site. We have corrected the pertinent sentence in the MS.

Unexpectedly, the divalent cation binding site is not occupied with the closest water (W739) at a distance of 1.7 Å from the site.

* Even if “in crystal” dimerization (via the crystallographic axis) can now be ruled out, are there any other crystal contacts that could otherwise preclude the protein to adopt a more relaxed/open configuration (typical of the inactive state)?

These would be crystal contacts either near the phosphorylatable Asp reaction center, or at the 4-5-5 face.

Phe670 is indeed adopting both active and inactive conformations; but further relaxation should for instance allow the a1-b1 loop to move away from the reaction center (e.g. Glu577 and Asp578 would normally move away from the site in the absence of phosphoryl-Mg²⁺ bound) etc Are there crystal contacts that would preclude these movements?

We have re-analyzed all-possible crystal contacts. No significant crystal contacts are observed either near the phosphorylatable Asp 623 site or at the 4-5-5 face. We have double-checked also any potential crystal contact near Glu577 and Asp578, but observed no crystal contact at all; thus, it can be ruled out that the conformation of the a1-b1 loop carrying these residues is affected.

7- It is very good to see that refinement/modeling improved substantially. If many of those strong Fourier difference peaks appear at the “missing” b4-b5 linker, I believe this deserves being described, for the readers to be aware.

Lines 91-92 should convey a better description, using some of the words the authors use now to respond to the reviewers (in the line of “There is density in this region, but not clear enough to model the linker segment reliably (Supp Fig X) ” etc). A supplementary panel will be helpful, showing the density, and hence enriching the image readers will get, beyond the simple absence of any model, which is what Fig 2a shows.

We agree with the reviewer and revised Fig S1 by showing electron density maps centered around the β 4 - β 5 linker region. We have added the relevant words in the text.

The structure is defined by continuous electron density from residues 570 to 689 except for residues 654 to 663, **with some poor density but not clear enough to model the linker segment reliably** (Supplementary Fig. 1).

8- Good to know that the I2 setting was well chosen and that the low number of ordered waters is probably due to the low solvent content.

Relevant literature is also better credited now, allowing to fit this nice piece of work into a broader context.

Glad these points have been settled.

We want to thank the referee very much for the careful work and valuable suggestions, which helped as we think to improve considerably the manuscript.